**EMBO**
Molecular Medicine

# MSTO1 is a cytoplasmic pro-mitochondrial fusion protein, whose mutation induces myopathy and ataxia in humans

Aniko Gal[1,2], Peter Balicza[2], David Weaver[1], Shamim Naghdi[1], Suresh K Joseph[1], Péter Várnai[3], Tibor Gyuris[4], Attila Horváth[4], Laszlo Nagy[4], Erin L Seifert[1], Maria Judit Molnar[2] & György Hajnóczky[1,*] (ID)

## Abstract

The protein MSTO1 has been localized to mitochondria and linked to mitochondrial morphology, but its specific role has remained unclear. We identified a c.22G > A (p.Val8Met) mutation of MSTO1 in patients with minor physical abnormalities, myopathy, ataxia, and neurodevelopmental impairments. Lactate stress test and myopathological results suggest mitochondrial dysfunction. In patient fibroblasts, MSTO1 mRNA and protein abundance are decreased, mitochondria display fragmentation, aggregation, and decreased network continuity and fusion activity. These characteristics can be reversed by genetic rescue. Short-term silencing of MSTO1 in HeLa cells reproduced the impairment of mitochondrial morphology and dynamics observed in the fibroblasts without damaging bioenergetics. At variance with a previous report, we find MSTO1 to be localized in the cytoplasmic area with limited colocalization with mitochondria. MSTO1 interacts with the fusion machinery as a soluble factor at the cytoplasm-mitochondrial outer membrane interface. After plasma membrane permeabilization, MSTO1 is released from the cells. Thus, an MSTO1 loss-of-function mutation is associated with a human disorder showing mitochondrial involvement. MSTO1 likely has a physiologically relevant role in mitochondrial morphogenesis by supporting mitochondrial fusion.

**Keywords** misato; mitochondria; mitochondrial disease; mitochondrial fusion; MSTO1

**Subject Categories** Genetics, Gene Therapy & Genetic Disease; Metabolism

## Introduction

Mitochondria are highly dynamic organelles whose morphology, distribution, and activity are dependent on fusion and fission (Chan, 2006). Several neurodegenerative diseases such as optic atrophy, Charcot–Marie–Tooth, Alzheimer's, Parkinson, and Huntington disease are associated with alterations of mitochondrial dynamics (Chan, 2006; Itoh *et al*, 2013).

MSTO1 (Misato) is an evolutionarily conserved nuclear DNA-encoded protein showing some structural similarity to eukaryotic tubulins and prokaryotic FtsZ. Null mutations of the *misato* in *Drosophila* are associated with irregular chromosomal segregation (Miklos *et al*, 1997; Gurvitz *et al*, 2002; Mottier-Pavie *et al*, 2011). However, depletion of MSTO1 by siRNA was also linked to mitochondrial fragmentation and cell death, and overexpressed EGFP-Misato was localized in the mitochondria and was shown to induce aggregation of mitochondria at the perinuclear region and cell death in HeLa cells (Kimura & Okano, 2007). In this study, MSTO1 was proposed to contribute to the cell's health as an outer mitochondrial membrane (OMM) resident affecting mitochondrial morphology. However, the mechanism by which MSTO1 supports mitochondrial morphogenesis and its potential human disease relevance remains elusive.

Disorders with primary mitochondrial dysfunction are clinically heterogeneous disorders that can manifest at any age, and can affect multiple tissues and organs, commonly the most energy demanding tissues, such as the central nervous system, skeletal and cardiac muscle, liver, endocrine glands, kidney, and the eyes. Mitochondrial diseases can result from mutations of genes located in either the nuclear DNA or the maternally inherited mitochondrial DNA. Until now, based on the relevant databases (Mitomap, MitoMiner, NCBI, Nexprot), approximately 300 different mtDNA pathogenic mutations have been found. The collection of nuclear genes encoding mitochondrial proteins is still changing, and only a few hundred

1  MitoCare Center for Mitochondrial Imaging Research and Diagnostics, Department of Pathology, Anatomy and Cell Biology, Thomas Jefferson University, Philadelphia, PA, USA
2  Institute of Genomic Medicine and Rare Disorders, Semmelweis University, Budapest, Hungary
3  Department of Physiology, Semmelweis University, Budapest, Hungary
4  Department of Biochemistry and Molecular Biology, University of Debrecen, Debrecen, Hungary
   *Corresponding author. Tel: +1 215 503 1427; E-mail: gyorgy.hajnoczky@jefferson.edu

disease-associated mutations of these genes have been identified. The minimum prevalence of all mitochondrial disorders is 1:2,000 (Suomalainen, 2015), but with the recent progress in identification of the mitochondrial proteins and the progress in genetic diagnosis, the number of human disease-associated mutations is expected to steeply increase.

In this study, we have identified an MSTO1 mutation in patients by whole-exome sequencing. This mutation segregated in the affected family and is absent in the other sequenced cases. Members of the affected family showed a diverse collection of the following alterations: severe myopathy, hypoacusis, endocrine dysfunctions, and psychiatric symptoms. These symptoms and laboratory tests indicate a possible role for the newly found *MSTO1* mutation in the background of mitochondrial disorders. Therefore, we have investigated mitochondrial dynamics and bioenergetics in both patient-derived cells and cell lines using genetic rescue and gene silencing, respectively. Collectively, our studies suggest that MSTO1 is a cytoplasmic protein required for mitochondrial fusion and network formation and its loss likely causes a multisystem disorder.

## Results

### Clinical data

#### Patient 1 (I/1) (Fig 1A)

A 53-year-old Hungarian female patient's symptoms started at the age of 38 with myalgia, weakness of the small hand muscles, and cognitive dysfunction. She was born as an immature, small baby from an overdue pregnancy. Her III digit on the foot was absent. Her motor and verbal development was delayed. Her bone age was also delayed. She had joint hyperlaxity and generalized lipomatosis. Presently, she has hyperthyroidism; mitral and tricuspidal insufficiency. Neurological examination revealed short stature (150 cm), micrognathia with small close-set eyes, myopia, myopathic face, bilateral hypoacusis, moderate atrophy of the small muscles of the hands and feet, pes varus. She had mild weakness in the distal muscles of the extremities. Deep tendon reflexes were decreased; pyramidal tract signs were not present. She had distal type hypaesthesia in the limbs. Mild truncal and upper limb ataxia and dysdiadochokinesis were present. She had anxiety and depressed mood.

Her brain MRI detected frontal atrophy and enlarged interhemispheric fissure and EMG showed myopathy. CK was in normal range. The resting serum lactate level was normal, while the lactate stress test indicated altered aerobic metabolism (resting lactate: 1.7 mmol/l (normal range: 1.0–2.0 mmol/l), after 15 min bicycling 0 min: 6.5 mmol/l, 5 min: 4.8 mmol/l, 15 min: 4.2 mmol/l, 30 min: 3.5 mmol/l (normal if the lactate level after 30 min exercise is less than twofold of the resting value)). Decreased vitamin D3 was measured (13.0 ng/ml, normal range 23.0–60.0 ng/ml). Using light microscopy, the myopathological investigations detected moderate muscle fiber caliber variation. No ragged blue or COX negative fibers were present. Electron microscopy found increased number of mitochondria and lipid droplets both in subsarcolemmal and intermyofibrillar localization. Many mitochondria had rounded shape; intramitochondrial paracrystalline inclusions were not present. Glycogen accumulation was also detected in these regions (Fig 1B).

#### Patient 2 (II/1)

The 30-year-old daughter of patient 1 (Fig 1A) was born from breech position. She was resuscitated. She had normal development. From age 11, episodically inflammatory lipomas occurred on her body. Her psychiatric symptoms stared at age 15 with depression and hallucinations. Severe undifferentiated type of schizophrenia was diagnosed. Some years later unilateral hypoacusis developed, and she had severe episodic ataxic gait repeatedly. Hyperthyreosis, hyperprolactanemia, and primer amenorrhea were detected as well. Neurological examination revealed mild weakness in the tibial anterior muscles and distal type hypaesthesia in the legs. Her CK and resting serum lactate level was in normal range. This patient did not agree to perform the lactate stress test. The vitamin D3 level was low (8.6 ng/ml). Her brain MRI detected pituitary adenoma and mild cerebellar ectopia.

#### Patient 3 (II/2)

The 24-year-old male patient was born from normal pregnancy. He had normal motor development. In his childhood, bone developmental problems were suspected based on his laboratory results (abnormal vitamin D3, calcium, phosphate, and ALP levels). The social anxiety started in the childhood. Later dyslexia, dysgraphia, dyscalculia, and learning difficulties were detected. His neurological examination revealed micrognathia, pectus excavatum, kyphoscoliosis, poor fine coordination, and slow psychomotility. CK was normal. Vitamin D3 was decreased (12.5 ng/ml).

#### Patient 4 (II/3)

The 20-year-old male patient was born as a hypotonic infant; his early development was delayed. He had a social anxiety since his early childhood and learning difficulties. Presently, he has autistic features with anxiety and impulsive behaviors. He had very long face, prominent jaw, and laxity of the knee joints. No neurological signs were found. The vitamin D3 level was low (10.8 ng/ml).

### Mutation analysis

Mitochondrial DNA mutation was not found in either blood or skeletal muscle samples, and single or multiple mtDNA deletions have been excluded as well.

Exome capture sequencing generated ~9.4 billion bases of sequence, and ~9.3 billion bases were then mapped to the target regions based on SeqCap_EZ_Exome_v3 Kit. 95% of the target regions had at least 10× coverage. After identification of variants, we focused only on non-synonymous variants, splice acceptor and donor site mutations, and short, frame shift coding insertions or deletions (indel). Using GEM.app software with autosomal dominant model after filtering out synonymous SNPs, 61 heterozygous variants remained, including 1 coding indel, 57 missense, 1 splice-site, 2 nonsense mutations. Further narrowing based on protein prediction scores resulted in 27 variations. We did not find any homozygous or compound heterozygous rare variants in the 1,015 genes, which were previously associated by the literature with mitochondrial function. The narrowing analysis for known mitochondrial genes filtered out 25 heterozygous (2 nonsense and 23 missense) variants. Among the rare variant filtering by the autosomal dominant inheritance model 16 genes were without any known function. Based on the literature, further seven genes were excluded because these genes are not linked to CNS involvements or mitochondrial dysfunctions.

   

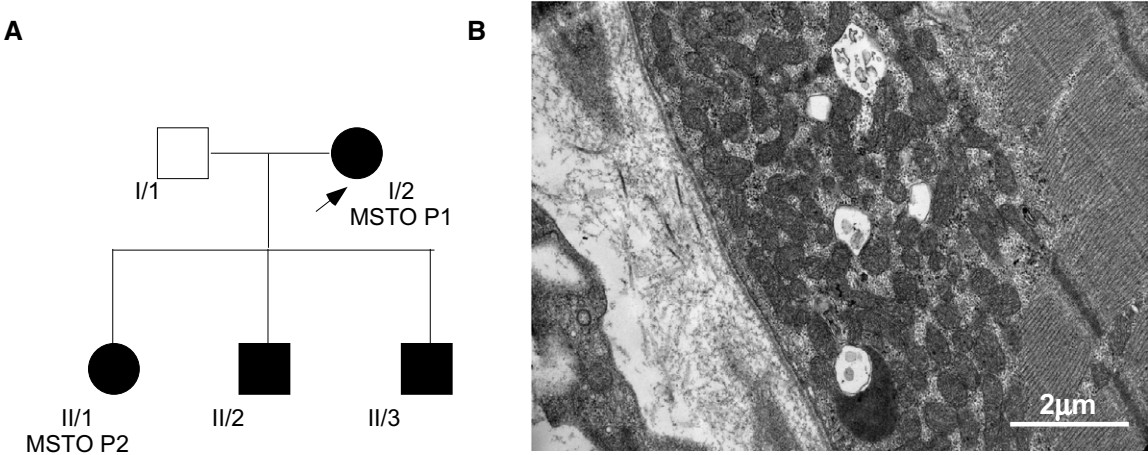

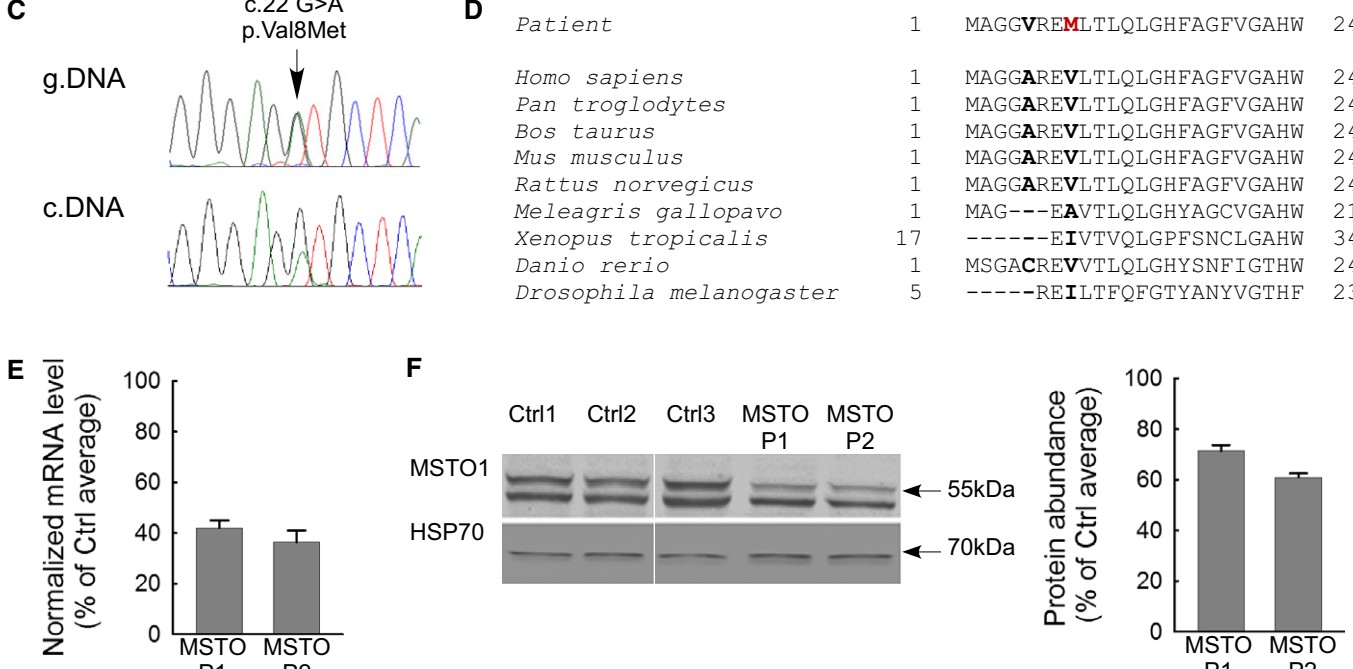

**Figure 1. Clinical and genetic data of the patient.**

A  Family tree of the investigated patients. Arrow indicates the proband.

B  Electron microscopy sections of the patient muscle biopsy specimen. Increased number of mitochondria both subsarcolemmal and intermyofibrillar, lipid droplets, and glycogen accumulation (electron microscopy, 30,000×).

C  Sequenogram of the suspected pathogenic mutation and the neighboring polymorphism in exon 1 of *MSTO1* gene from genomic (upper part) and cDNA (lower part). Arrow indicates the position of the mutation.

D  Taxonomical alignments of the affected MSTO1 protein sequence. Location of the alterations in the patients are shown in bold. The red "M" indicates the amino acid substitution segregated in all affected family members.

E  Normalized mRNA expression level from the patient primary fibroblasts (percentage of the average value of the healthy controls) (mean ± SEM).

F  MSTO1 Western blotting of the patient and control fibroblast. Left: representative blots; right: normalized protein abundance of the percentage of the average protein expression levels of the controls (mean ± SEM).

Source data are available online for this figure.

Finally, four missense mutations were validated with Sanger sequencing. Among them only the c.22 G>A (p.Val8Met) substitution in *MSTO1* gene is segregated in all affected family members and was present in heterozygous form (Table EV1 and Fig 1C). This mutation was found in urinary tract and colorectal tumors, as a somatic mutation (COSM3930426, COSM3930426) (http://cancer.sa

nger.ac.uk); according to the Exome Aggregation Consortium (ExAC) database (http://exac.broadinstitute.org), the minor allele frequency is 0.003% (rs762798018), and it was absent in 1000 Genome (http://www.1000genomes.org), NHLBI Exome Sequencing Project (ESP) (http://evs.gs.washington.edu/EVS/), ClinVar (http://www.ncbi.nlm.nih.gov/clinvar), dbGAP (http://www.ncbi.nlm.nih.gov/gap), and EGA (http://www.ebi.ac.uk/ega) databases. Connection with any clinical phenotype has not been described, yet. The mutated part of MSTO1 protein sequence is highly conserved in mammals (Fig 1D). This alteration was confirmed by cDNA sequencing from fibroblast as well (Fig 1C). Other alterations of *MSTO1* gene were excluded by Sanger sequencing of the total coding sequence from genomic DNA and cDNA sequencing from patient derivate fibroblasts (MSTO P1, II/1 and MSTO P2, I/2). The copy number alteration was also excluded by real-time PCR methodology.

In the patient-derived primary fibroblast culture, the MSTO1 mRNA and protein expression were significantly decreased (MSTO P1 and MSTO P2) compared with the average values of three controls (Fig 1E and F). The MSTO1 mRNA expression was $42.0 \pm 3.0\%$ in MSTO P1 and $36.3 \pm 4.7\%$ in MSTO P2 (Fig 1E), while the protein abundance was $71.4 \pm 2.3\%$ in MSTO P1 and $61.0 \pm 1.6\%$ in MSTO P2 (Fig 1F). The other two affected family members did not agree to the skin biopsy.

### *In silico* analysis

Based on the prediction of the InterPro domain software, MSTO1 protein has 2 tubulin/Ftz-like GTPase domains. The prediction of GTP binding residues in the protein sequence by GTP-binder application (Chauhan *et al*, 2010) found seven possible GTP binding sites in the first (aa 16-21; 23-24; 30; 62-63; 73-74; 76-77; and 81) tubulin/Ftz GTPase domain (Fig EV1A).

The protein alignments between MSTO1 and the known mitochondrial fusion proteins detected some similarities in a 12 amino acid span with the OMM fusion proteins, MFN1 and MFN2 (Fig EV1B). These amino acids are located in the second tubulin/Ftz, GTPase domain of the MSTO1 protein and are highly conserved in mammals (Fig EV1C). To determine the significance of the similarity between MSTO1 and MFN1, BLAST alignments using randomly scrambled versions of each protein were performed with the default parameters for multiple sequence alignment (word size 3, BLOSUM62 matrix). Out of 100 alignments MFN1 and scrambled MSTO1s and 100 alignments of MSTO1 and scrambled MFN1s, 17 (8.5%) aligned segments with an equal or higher bit-score were found. Thus, the homology is not statistically significant in and of itself, though we would note the strong conservation across species of the acidic amino acids in MSTO1 and mitofusins. No similarity was found between MSTO1 and the inner mitochondrial membrane (IMM) fusion protein, OPA1.

### Mitochondrial morphology in primary fibroblasts

Mitochondrial morphology was visualized in fibroblasts obtained from I/2: MSTO1 Patient 1 (P1) and II/1: MSTO1 Patient 2 (P2), the 2 family members who agreed to undergo skin biopsy, and from 3 individuals serving as controls (Ctrl 1, 2, and 3), expressing mitochondrial matrix-targeted DsRed1 (mtDsRed). The cells with various mitochondrial morphology (Fig EV2A) were quantified as a percentage of the total number of transfected cells ($\geq$ 20 cells per experiment, $n = 5$ independent experiments). More cells with aggregated mitochondria were counted in MSTO1 patient fibroblasts than in the controls (P1: 37.8%; P2: 33.3%; Ctrl average: $11.7 \pm 0.4\%$). In patient fibroblasts, the percentage of cells with fragmented and partially fragmented mitochondria was also increased (P1, partially fragmented: 46.7%, fragmented: 17.3; P2: partly fragmented: 48.8%, fragmented: 8.5%; Ctrl average: partly fragmented: $25.1 \pm 1\%$, fragmented: $0 \pm 0\%$) (Fig EV2A–C).

### Mitochondrial continuity in primary fibroblasts

To evaluate whether the distinct mitochondrial morphology in patient and control fibroblasts can result from different levels of interorganellar continuity and fusion events, the cells were co-transfected with cDNA encoding mtDsRed1 and mitochondrial matrix-targeted photoactivatable GFP (mtPA-GFP). Using confocal microscopy, time series of fluorescence images were recorded and 25 μm² square-shaped areas were illuminated by a pulsed laser to photoactivate mtPA-GFP (Eisner *et al*, 2014; Weaver *et al*, 2014), which is a soluble protein that shows rapid diffusion (Partikian *et al*, 1998). Mitochondrial matrix continuity and connectivity is unveiled by the diffusion of the photoactivated mtPA-GFP to the regions outside the 2P illuminated area. Image time series shows that mtPA-GFP diffusion was slower in P1 than in the age-matched Ctrl3 fibroblasts (Fig 2A). To quantify connectivity, the time course of the ratio of $F_{mtPA\text{-}GFP}$ and $F_{mtDsRed}$ was calculated for the region of photoactivation (RPA) in both P1 and P2 and in the three controls (Fig 2B). At 500 s, the lesser decay of the fluorescence ratio in the RPA indicates a decrease in the combined activity of mitochondrial network formation, mitochondrial fusion, and mitochondrial movements in the MSTO1 patient cells (Fig 2B).

### Mitochondrial fusion dynamics in primary fibroblasts

To validate fusion events, reciprocal spreading of mtPA-GFP and mtDsRed among mitochondria that were not continuous at the time of mtPA-GFP photoactivation was sought (Fig 3A). The fusion events were quantitatively analyzed by the progression of the distribution of mtPA-GFP fluorescence between the images collected in the first 8 min after photoactivation. The result of manual counting indicates less mitochondrial fusion activity in the MSTO1 patient fibroblasts (Fig 3B). The total fusion numbers were decreased significantly in the patient fibroblasts (P1: $4.01 \pm 0.13$ fusion/run; P2: $3.50 \pm 0.12$ fusion/run; Ctrl average: $6.39 \pm 0.24$ fusion/run) (Fig 3B). In a previous study, we have quantified the effect of two different ADOA-associated OPA1 mutations on the mitochondrial fusion activity using the same manual counting method used in the present study (Eisner *et al*, 2014). G300E point mutation in the GTPase domain and Δ58 deletion of the GTPase domain caused 60–70% decrease in the fusion activity, whereas the present MSTO1 mutations induced an approximately 40% decrease in fusion activity. However, we have also studied an ADOA-associated OPA1 mutation (c.984) that caused a lesser decrease in the fusion activity (30%) than the present MSTO1 mutations. Thus, the MSTO1-associated fusion decrease is in the range of the OPA1-associated ones.

Fusion activity was also quantified by an algorithm that calculates the loss of the green-only pixels during the

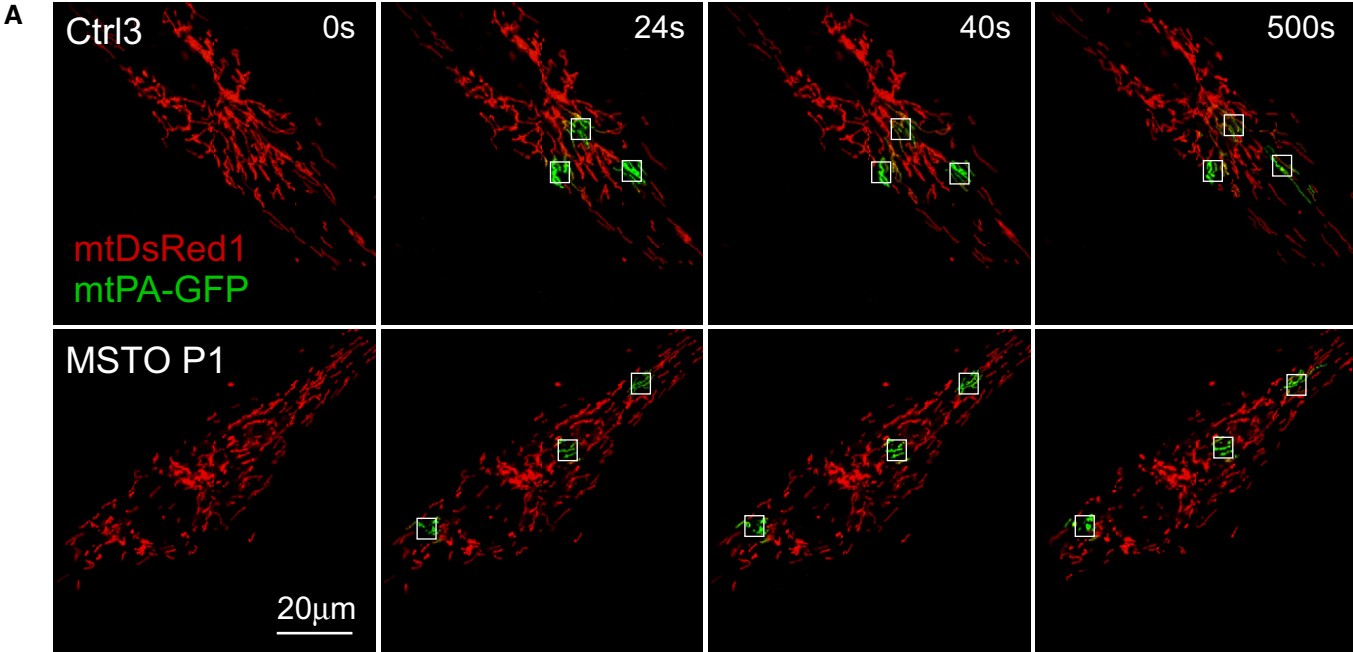

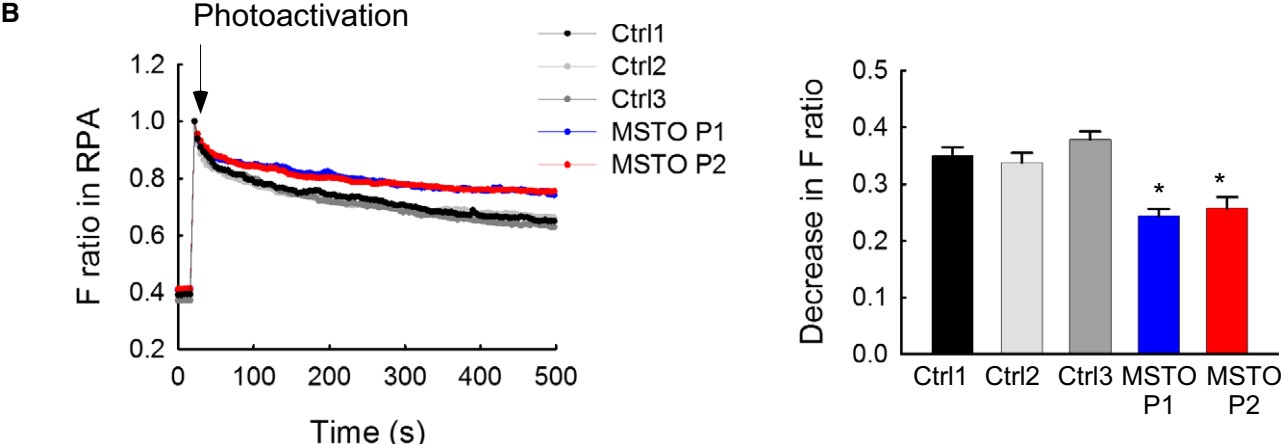

**Figure 2.  Mitochondrial continuity in primary fibroblasts.**

A   Image time series showing a representative fibroblast cell before and after 2P photoactivation of mtPA-GFP (white squares): 0 s (pre-activation), 24 s (immediately after photoactivation), 40 s, and 500 s.

B   The time course of the ratio of $F_{mtPA\text{-}GFP}$ to $F_{mtDsRed}$ for the region of photoactivation (RPA) (left); the decay of the fluorescence ratio in the RPA at 500 s (right) (number of imaged cells: Ctrl1 $N$ = 94; Ctrl2 $N$ = 45; Ctrl3 $N$ = 87; MSTO P1 $N$ = 84; MSTO P2 $N$ = 130 from minimum three experiments per each cells). Stars indicate the significant differences (Student's $t$-test: Ctrl average vs. MSTO P1 *$P$ = 1.44 × $10^{-9}$, Ctrl average vs. MSTO P2 *$P$ = 1.52 × $10^{-12}$; one-way ANOVA: $P$ = 0.00815) (mean ± SEM).

post-photoactivation period. This unbiased determination showed that the average half-time of the GFP-only pixel loss was increased significantly in the MSTO1 fibroblasts (P1: 1,914 ± 305 s; P2: 1,817 ± 312 s; Ctrl average: 713 ± 105 s). Thus, results obtained by two different approaches show that mitochondrial fusion activity is suppressed in the MSTO1-deficient patient cells.

The average duration of individual fusion events was similar in both MSTO1 fibroblasts and in age-matched controls (P1: 133 ± 7 s; P2: 133 ± 7 s; Ctrl average: 117 ± 6 s) (Fig 3C). Furthermore, the distribution of different fusion types in terms of rapid reversal (Fig 3D) and orientations (Fig 3E) did not show any alterations in

MSTO1 fibroblasts as compared with controls. In addition, the abundance of the main fusion (MFN1, MFN2, and OPA1) and fission (DRP1) proteins was unaltered in MSTO1 fibroblasts (Fig 3F).

**MSTO1 overexpression restores mitochondrial fusion dynamics in MSTO1 patient fibroblasts**

To test whether normal mitochondrial fusion phenotype can be restored by MSTO1 overexpression in MSTO1 patient fibroblasts, a bicistronic expression vector for MSTO1 and mtDsRed was created (mtDsRed1-T2A-MSTO1). Upon transfection of the P2 fibroblasts

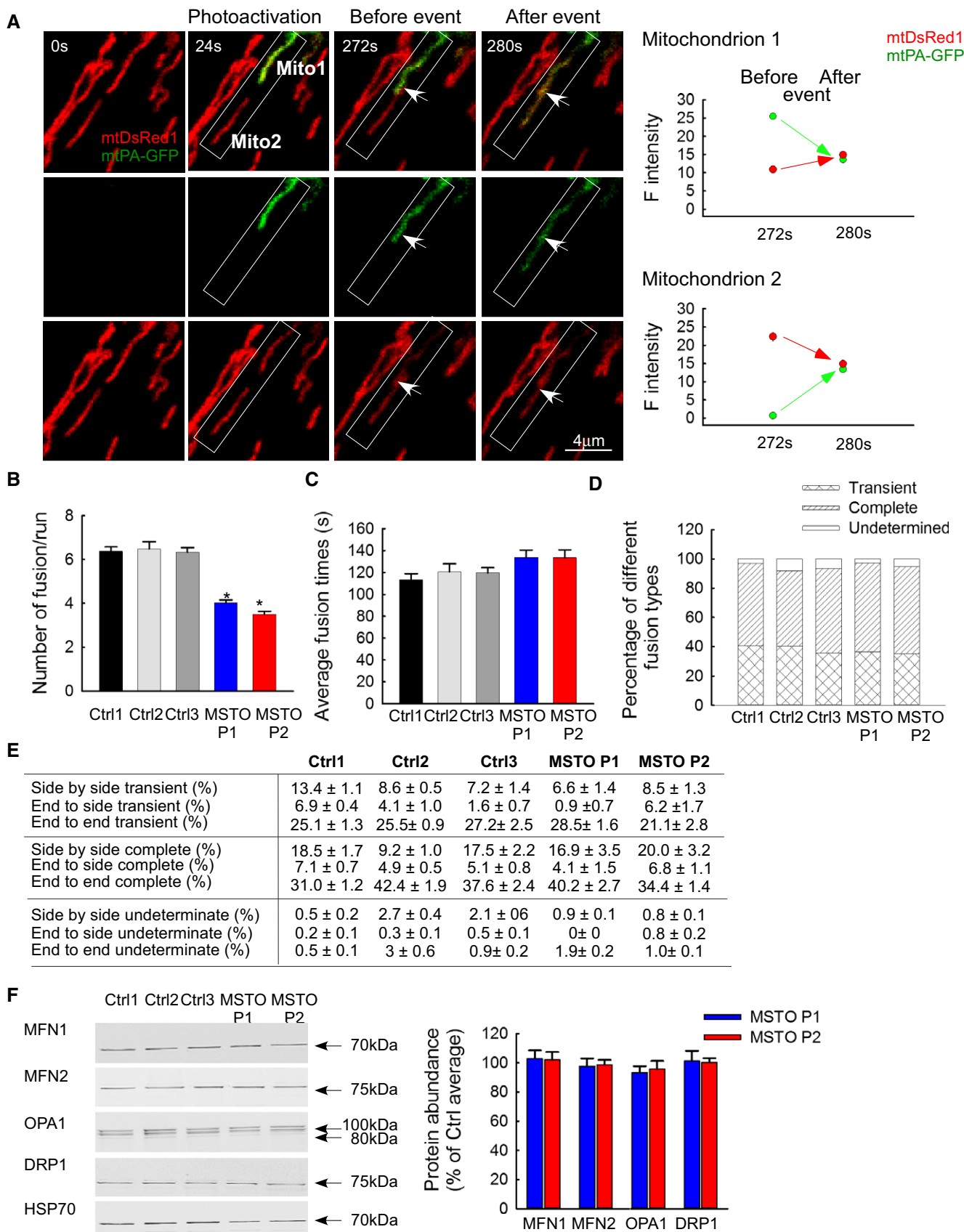

**Figure 3.**

**Figure 3.  Mitochondrial fusion events in primary fibroblast.**

A   Representative example is shown out of 2,200 events. During photoactivation of mtPA-GFP, mtDsRed is photobleached, allowing detection of the mixing of each fluorescent protein after mitochondrial fusion. Mitochondria shown in the white box undergo an end-to-end fusion event at 272 s. During the event, mitochondrion #1 donates fluorescent mtPA-GFP to mitochondrion #2 (acceptor). At the same time, the green acceptor mitochondrion donates fluorescent mtDsRed to mitochondrion #1. Arrows show the location of the fusion, abrupt, and complementary inter-mitochondrial transfer of the fluorescent proteins. The graphs on the right show the average mtDsRed1 and mtPA-GFP fluorescence intensities of mitochondrion 1 and mitochondrion 2 before and after fusion event. The white box indicates two mitochondria before and after fusion.

B   Rates of fusion events in primary fibroblasts. Stars indicate the significant differences (Student's $t$-test: Ctrl average vs. MSTO P1 *$P = 9.6 \times 10^{-21}$, Ctrl average vs. MSTO P2 *$P = 5.3 \times 10^{-31}$; one-way ANOVA: $P < 0.00001$) (mean $\pm$ SEM).

C   Average duration of fusion events which were followed by fission during the 8-min recording time (mean $\pm$ SEM).

D   Distribution of transient and complete type fusion events.

E   Orientation characteristics of fusion events in different fibroblasts (mean $\pm$ SEM).

F   Western blot of the main fusion and fission proteins. Left: representative blots; right: quantified protein abundances in MSTO cells relative to the control cell average (mean $\pm$ SEM).

Data information: (B–E) Number of fusions: Ctrl1 $N$ = 599; Ctrl2 $N$ = 292; Ctrl3 $N$ = 551; MSTO P1 $N$ = 299; MSTO P2 $N$ = 459; number of imaged cells: Ctrl1 $N$ = 94; Ctrl2 $N$ = 45; Ctrl3 $N$ = 87; MSTO P1 $N$ = 84; MSTO P2 $N$ = 130 from minimum three experiments per each cells.
Source data are available online for this figure.

with this vector, the abundance of MSTO1 protein was greatly increased as compared with pcDNA-transfected cells (186%, Fig 4A). Only a hardly noticeable band appeared at the expected molecular weight of the mtDsRed1-T2A-MSTO1 complex, indicating that it was effectively processed to give rise to separated MSTO1 and mtDsRed proteins (Fig 4A).

To evaluate fusion dynamics, the fibroblasts were then co-transfected with mtPA-GFP and either mtDsRed1-T2A-MSTO1 or mtDsRed and pcDNA. In the microscopy studies, the MSTO1 overexpressing cells were identified based on the mtDsRed fluorescence (images in Fig 4B). The spreading of mtPA-GFP fluorescence from the areas of photoactivation was accelerated in the MSTO1 overexpressing cells (Fig 4B). The decrease in the fluorescence ratio at the RPA was significantly greater after MSTO1 transfection and was similar to the average value of the three controls (MSTO1-transfected P2 cells: $0.37 \pm 0.02$; only mtDsRed-transfected MSTO P2 cells: $0.2 \pm 0.01$; average of controls: $0.35 \pm 0.02$ s) (Fig 4C). The average half-time of the GFP-only pixel loss was decreased significantly in the MSTO1 overexpressed cells (pcDNA-transfected cells: $1,714 \pm 300$ s, MSTO1-transfected cells: $954 \pm 137$ s, $n_{pcDNA} = 34$; $n_{MSTO1} = 36$; $P = 0.00343$). The MSTO1 gene delivery normalized the fusion numbers as well (Fig 4D). After MSTO1 transfection, the average fusion number was $5.2 \pm 0.4$/run ($0.65 \pm 0.0$/min), while in the mtDsRed-transfected cells $3.52 \pm 0.3$/run ($0.44 \pm 0.02$/min) fusion events were counted (Fig 4D). As a reference, the average fusion numbers of control fibroblasts were $6.4 \pm 0.1$/run, $0.8 \pm 0.0$/min (Fig 4D). However, MSTO1 overexpression did not alter the reversibility (Fig 4E), duration (Fig 4F), and orientation (Fig 4G) of the fusion events.

**MSTO1 silencing inhibits mitochondrial fusion in HeLa cells**

Next, in order to avoid the complexities and high variability of the cells derived from different individuals, the possible contribution of MSTO1 to fusion dynamics was assessed in HeLa cells. Gene silencing was performed with either MSTO1-targeting or scrambled siRNA. Immunoblotting confirmed the MSTO1 depletion (Fig 5A). Following 72-h silencing, the protein amount of MSTO1 was $49 \pm 3$% compared to the scrambled siRNA control. In the MSTO1-silenced cells, the expression of MFN1, MFN2, OPA1, and DRP1 was not significantly changed (Fig 5A).

Mitochondrial fusion dynamics was evaluated upon co-transfection of the siRNA-treated cells with mtPA-GFP and mtDsRed. The siMSTO1 cells showed increased mitochondrial fragmentation (Fig EV2D and E) and relatively slow spreading of mtPA-GFP from the photoactivation areas (Fig 5B and C). Applying the fusion quantification algorithm, the average half-time of the GFP-only pixel loss, which inversely correlates with mitochondrial fusion activities, was increased significantly in the MSTO1-silenced cells (MSTO1-silenced cells: $505 \pm 101$ s, scramble siRNA-silenced cells: $247 \pm 20$ s, $n_{Scr} = 45$; $n_{MSTO1} = 58$; $P < 0.00001$). Consistent with this, the manually counted fusion number also decreased in the siMSTO1 condition (Fig 5D). After MSTO1 siRNA transfection, the average fusion number was $10.9 \pm 0.1$/run, while in the scramble siRNA-transfected cells $20.1 \pm 0.5$/run fusion events were counted (Fig 5D). The average duration of the fusion events slightly increased upon MSTO1 targeting (Fig 5E), whereas the distribution of the different fusion types (Fig 5F) and the orientation of the fusion events (Fig 5G) were not changed.

**Mitochondrial motility is maintained in patient fibroblasts**

We have previously shown that mitochondrial fusion can be controlled indirectly through changes in mitochondrial motility (Liu et al, 2009; Twig et al, 2010). Therefore, mitochondrial motility was quantified in the MSTO1 patient fibroblasts as well as in the controls in side-by-side comparisons (Fig EV3). The results show that P2 had similar mitochondrial movement activity to Ctrl1 and more than Ctrl2 ($P = 0.97$ and $P = 0.0063$, respectively, Kruskal–Wallis one-way ANOVA, Dunn's multiple comparison test), and P1 was similar to Ctrl3 in terms of mitochondrial motility ($P = 0.10$, Mann–Whitney rank sum test). Thus, the decrease in mitochondrial fusion is not secondary to a decrease in mitochondrial motility in the MSTO1-deficient patient cells.

**Mitochondrial membrane potential, $Ca^{2+}$ uptake, and respiration after MSTO1 silencing**

To determine whether the effect of MSTO1 silencing on mitochondrial fusion might be secondary to an effect on mitochondrial bioenergetics, the mitochondrial membrane potential ($\Delta\Psi_m$) and $Ca^{2+}$ uptake were determined. The resting $\Delta\Psi_m$ and the $\Delta\Psi_m$ loss evoked by $Ca^{2+}$ additions were similar in both MSTO1- and scramble siRNA-silenced cells (Fig 6A). The resting "cytoplasmic" $[Ca^{2+}]$ and

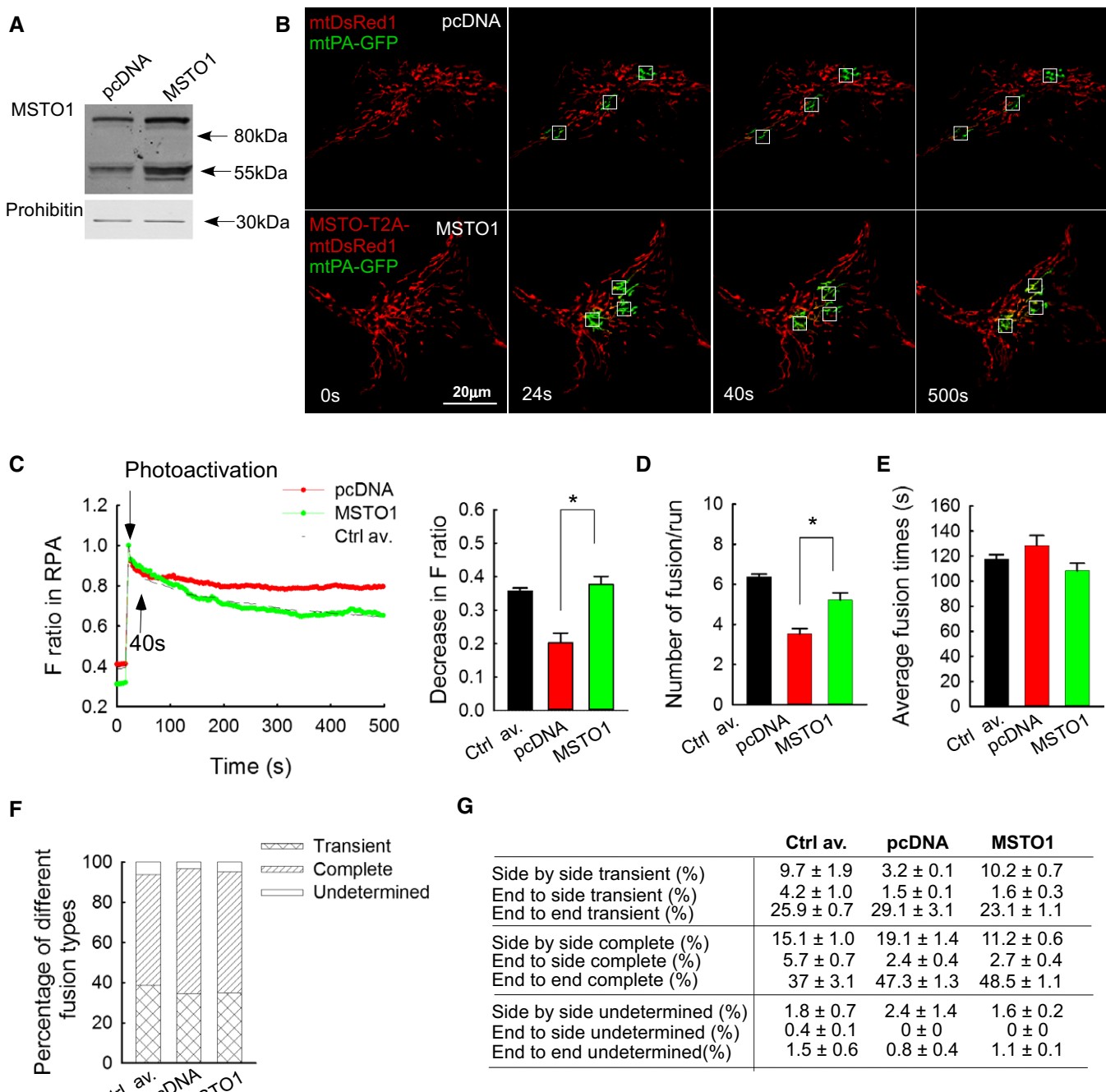

**Figure 4.  MSTO1 overexpression in MSTO1 patient fibroblasts.**

A   Representative Western blot of MSTO1 protein expression after mtDsRed1-T2A-MSTO1 vector delivery (representative of three experiments).

B   Image time series showing representative pcDNA- and mtDsRed1-T2A-MSTO1-transfected fibroblast cell before and after 2P photoactivation of mtPA-GFP (white squares) as in Fig 2A.

C   The time course of the ratio of $F_{mtPA-GFP}$ to $F_{mtDsRed}$ for the region of photoactivation (RPA) (left); the decay of the fluorescence ratio in the RPA at 500 s (right). Stars indicate the significant differences (Student's t-test: *$P$ = 0.00412, ANOVA: $P$ = 0.00859) (mean ± SEM).

D   Rates of fusion events in primary fibroblasts. Stars indicate the significant differences (Student's t-test: *$P$ = 1.71 × 10$^{-4}$, ANOVA: $P$ = 0.00343) (mean ± SEM).

E   Average duration of fusion events which were followed by fission during the 8-min recording time (mean ± SEM).

F   Distribution of transient and complete type fusion events.

G   Orientation characteristics of fusion events in different fibroblasts (mean ± SEM).

Data information: (C–G) Number of fusions: mtDsRed1-T2A-MSTO1: $N$ = 188; pcDNA + DsRed $N$ = 127; number of imaged cells: mtDsRed1-T2A-MSTO1: 35; pcDNA + DsRed $N$ = 36; from ≥ 2 experiments per each cells.

Source data are available online for this figure.

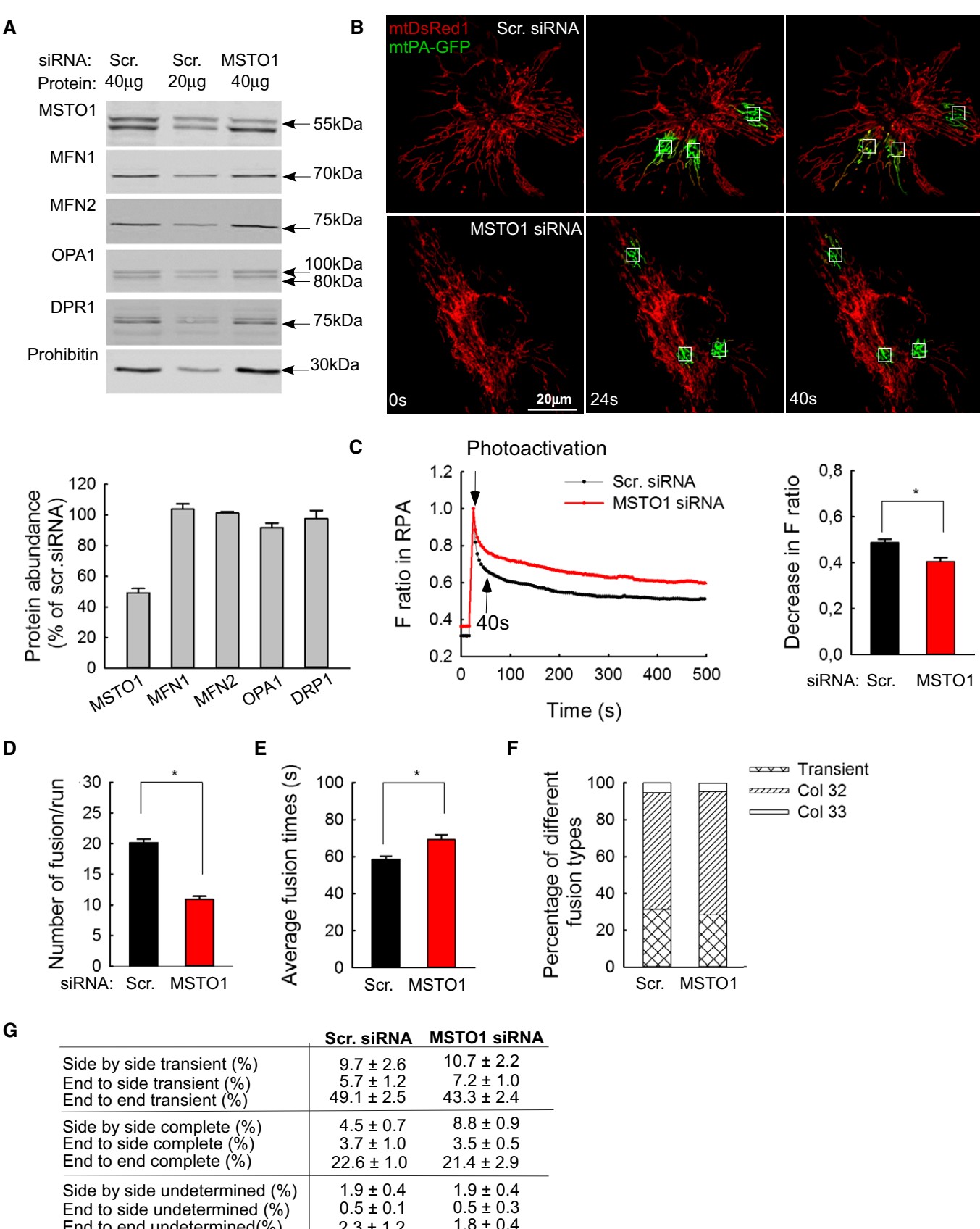

**Figure 5.**

**Figure 5.  MSTO1 silencing in HeLa cells.**

A  Protein expression of MSTO1 and the main fusion and fission proteins after MSTO1 silencing. Top: representative blots; bottom: normalized protein abundance of the percentage of the average protein expression levels of the scramble siRNA-treated cells ($N = 3$) (mean $\pm$ SEM).

B  Image time series showing a representative MSTO1- and scramble siRNA-silenced HeLa cell before and after 2P photoactivation of mtPA-GFP (white squares) as in Fig 2A.

C  The time course of the ratio of $F_{mtPA\text{-}GFP}$ to $F_{mtDsRed}$ for the region of photoactivation (RPA) (left); the decay of the fluorescence ratio in the RPA at 500 s (right). Stars indicate the significant differences (Student's $t$-test: $*P = 5.4 \times 10^{-4}$, one-way ANOVA: $P = 0.03678$) (mean $\pm$ SEM).

D  Rates of fusion events in silenced HeLa cells. Stars indicate the significant differences (Student's $t$-test: $*P = 7.15 \times 10^{-19}$, one-way ANOVA: $P < 0.00001$) (mean $\pm$ SEM).

E  Average duration of fusion events which were followed by fission during the 8-min recording time. Stars indicate the significant differences (Student's $t$-test: $*P = 3.67 \times 10^{-4}$, one-way ANOVA: $P = 0.00052$) (mean $\pm$ SEM).

F  Distribution of transient and complete type fusion events.

G  Orientation characteristics of fusion events in silenced and control HeLa cells (mean $\pm$ SEM).

Data information: (C–G) Number of fusions: MSTO1 siRNA $N = 386$; Scr. siRNA $N = 563$; number of imaged cells: MSTO1 siRNA $N = 54$; Scr. siRNA $N = 50$; from three experiments.

Source data are available online for this figure.

the transients caused by $CaCl_2$ boluses were also similar in the MSTO- and scrambled siRNA-treated cells (Fig 6B).

Respiration function measurements showed that MSTO silencing was without major effect on basal (Fig 6C) or leak-dependent $JO_2$, but led to a slight but significant lowering of the maximal electron transport chain activity (Fig 6C) as determined by injecting the chemical uncoupler FCCP.

Thus, $\Delta\Psi_m$ generation, mitochondrial $Ca^{2+}$ accumulation and basal respiration do not seem to be affected by short-term MSTO1 depletion, indicating that suppression of mitochondrial fusion is not secondary to a change in mitochondrial bioenergetics.

**MSTO1 protein is mainly localized in the cytoplasmic area and shows partial colocalization with the mitochondria**

In HeLa cells, overexpression of either MSTO1-Myc or mtDsRed1-T2A-MSTO1 caused an increase in MSTO1 protein amount (465 and 542%, respectively, compared to that in the pcDNA-transfected cells, Fig 7A). Again, the 80 kDa band representing the uncleaved mtDsRed1-T2A-MSTO1 complex was faint (Fig 7A).

To test the subcellular distribution of MSTO1, immunostaining was performed. HeLa cells transfected with mtDsRed1-T2A-MSTO1 vector were stained with anti-MSTO1 antibody 24 h after transfection. MSTO1 did not show colocalization with the mitochondria; rather, intense MSTO1 staining was present in the whole cytoplasmic area (Fig 7B). To further investigate the localization of MSTO1 protein, HeLa cells were co-transfected with OMM-targeted GFP (OMP25-GFP) and MSTO-cMyc vectors. 24 h after transfection, the coverslips were divided into two groups. One group was permeabilized with digitonin (30 μg/ml) before fixation, while the other group was just fixed. Both the intact and permeabilized cells were stained with anti-MSTO1 or anti-cMyc antibodies. In the intact cells, the MSTO1/Myc signal and mitochondrial targeted OMP25-GFP colocalization was approx. 30% (MSTO1 staining: $29 \pm 2.7\%$; c-Myc staining: $33.1 \pm 2.1\%$) (Fig 7C and D). In the permeabilized cells, most of the MSTO1/c-Myc immunostaining disappeared, and for the remaining signal, the colocalization was around 50% (MSTO1 staining: $56 \pm 1.2\%$; c-Myc staining: $48.2 \pm 3.1\%$) (Fig 7C lower images; and Fig 7E).

To complement these results, endogenous MSTO1 protein abundance was assessed by immunoblotting in both total cell lysate and mitochondria isolated from HeLa cells. When normalized to the amount of the mitochondrial loading control, prohibitin, the MSTO1 protein amount was relatively low, only 35% in the mitochondrial fraction (Fig 7F).

**MSTO1 protein is soluble**

Permeabilization of the plasma membrane resulted in almost complete loss of the immunofluorescence, whereas the mitochondrial markers stayed (Fig 7C), indicating that MSTO1 cannot be an integral or firmly associated mitochondrial protein. Then, it became important to determine how this protein could interact with the mitochondrial fusion proteins, which are transmembrane components of the OMM or IMM. Therefore, we compared the release upon plasma membrane permeabilization of MSTO1 to other cellular proteins (Fig 8A). After plasma membrane permeabilization, in both the MSTO1-transfected and non-transfected HeLa cells, the MSTO1 protein is promptly and almost completely released into the cytoplasmic fraction. In MSTO1 transfected cells after 140 s more than 60% of MSTO1 protein was released into the cytoplasmicfraction, while in the non-transfected cells MSTO1 release was even higher, approximately 88% (Fig 8B). For comparison, we tested the release of proteins with known localization. Hexokinase II, known to be loosely bound to the OMM (Chiara *et al*, 2008), showed less and slower release than MSTO1 in both conditions (Fig 8B). The α-tubulin, a soluble cytosolic protein, showed a very similar release pattern to MSTO1 (Fig 8B). After saponin treatment, in the control and MSTO1 primary fibroblast cells, similar protein releases were found.

Mitochondrial intermembrane space proteins represented by cytochrome c were not released following digitonin or saponin treatments, since the OMM is more resistant to these detergents than the plasma membrane. However, cytochrome c redistributed to the cytoplasmic fraction upon tBid (pro-apoptotic Bcl2 protein) addition (Willis *et al*, 2007). Lastly, permeabilized cells were exposed to limited proteolysis with trypsin (100 μg/ml), followed by addition of soybean trypsin inhibitor (250 μg/ml) (Fig 8). After trypsin treatment, the MSTO, like hexokinase or α-tubulin, disappeared from both the cytoplasmic and membrane fractions, which further indicated that MSTO1 is localized out of the mitochondria and it can be loosely bound to the OMM.

## Discussion

In this report, we described the first identified human disease-associated variation of *MSTO1* gene in a patient with multisystemic clinical phenotype. Furthermore, we demonstrated that MSTO1 has

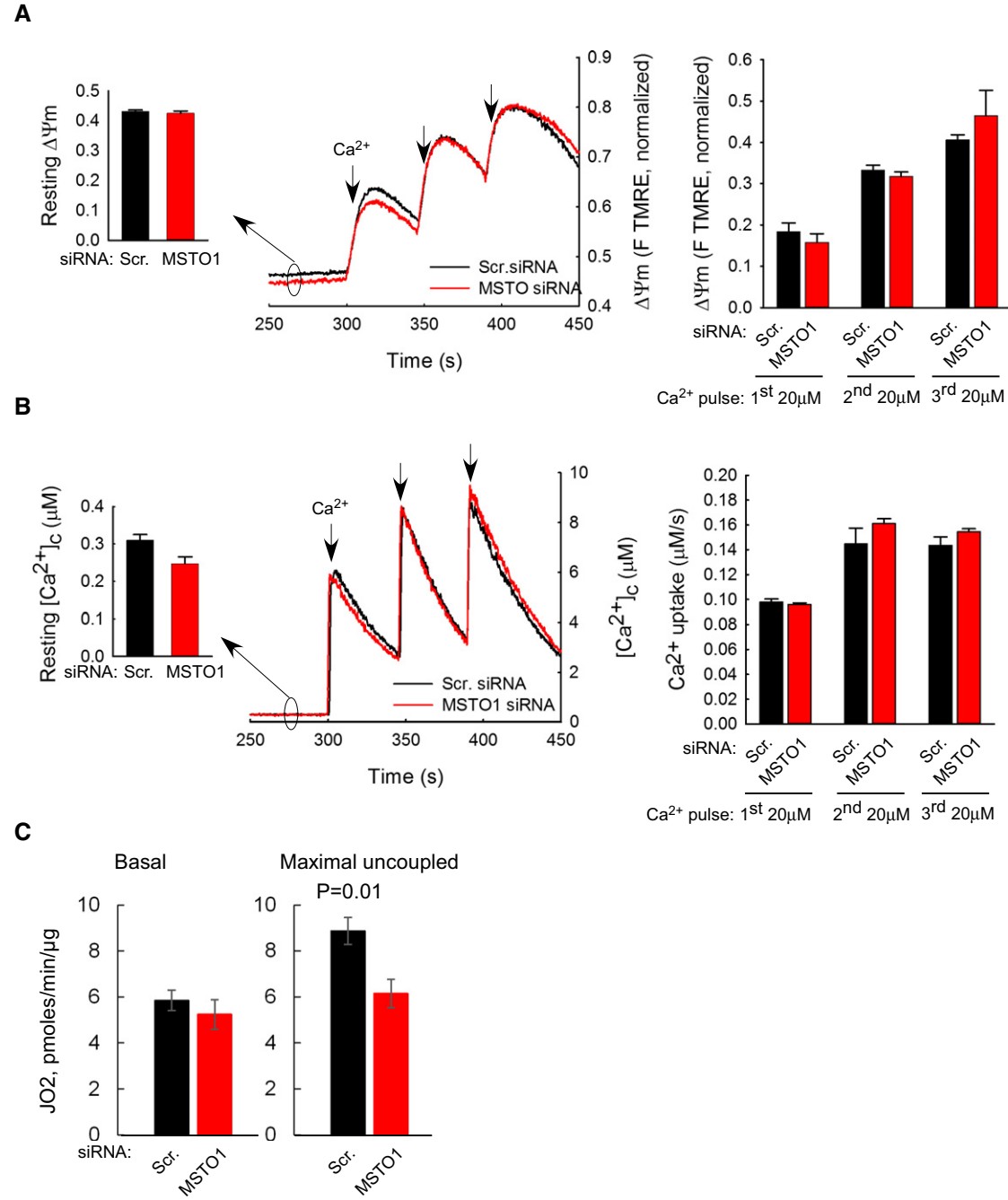

**Figure 6.  Mitochondrial membrane potential and mitochondrial Ca²⁺ uptake after MSTO1 silencing.**

A  $\Delta\Psi_m$ measured using TMRE in suspensions of permeabilized HeLa cells. Increase in fluorescence after each CaCl₂ pulse indicates depolarization. Left inset shows resting $\Delta\Psi_m$ as initial TMRE fluorescence normalized to the fluorescence attained after complete depolarization by uncoupler (5 μM FCCP). Left graphs shows the $\Delta\Psi_m$ loss evoked by Ca²⁺ pulses (4 and 3 pairs of scrambled and MSTO siRNA-silenced cells per measurement from two independent experiments). Arrows indicate the CaCl₂ additions (20 μM each) (mean ± SEM).

B  Clearance of Ca²⁺ by mitochondria. The cells were challenged by repetitive CaCl₂ pulses (20 μM each). Ca²⁺ uptake is shown in the representative traces by the decay of the [Ca²⁺] increases evoked by each CaCl₂ addition (arrows). Left inset shows the mean resting "cytoplasmic" Ca²⁺ concentrations, the left graph shows mitochondrial Ca²⁺ uptake after each Ca²⁺ pulse (4 and 3 pairs of scrambled and MSTO siRNA-silenced cells per measurement from two independent experiments) (mean ± SEM).

C  JO₂ was measured using the Seahorse XF24 analyzer and calculated by subtracting the O₂ consumption rates measured prior to injecting the complex III inhibitor antimycin A (1 μM) from the rate measured after antimycin addition. Basal JO₂ refers to the JO₂ in cells exposed only to DMEM + substrates. Maximal JO₂ refers to the JO₂ after injecting the chemical uncoupler FCCP (250 nM). *P*-value was calculated using Student's *t*-test ($P = 0.01$). *n* = 5 different passages per siRNA treatment (mean ± SEM).

Source data are available online for this figure.

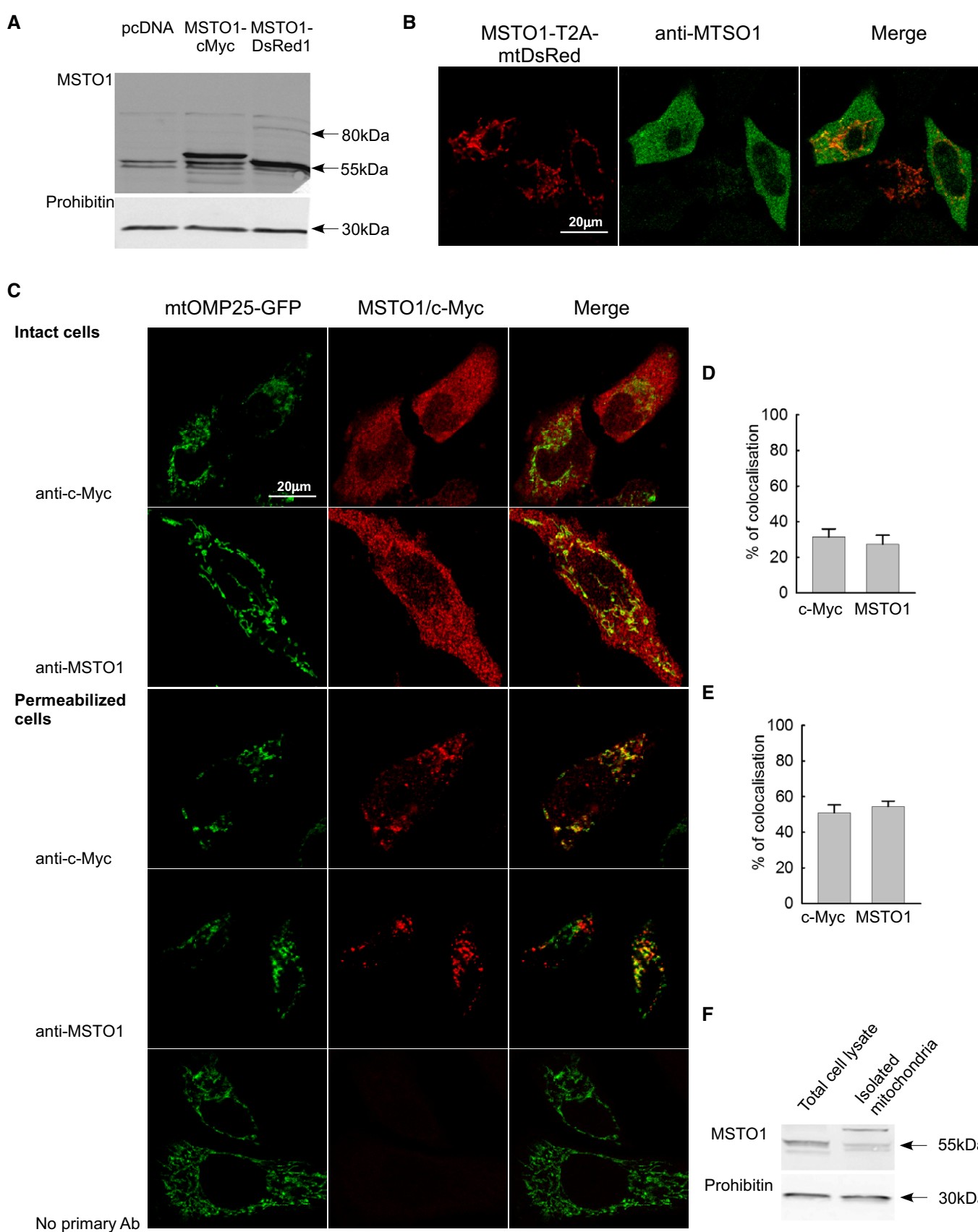

**Figure 7.**

◀

**Figure 7.  Subcellular localization of MSTO1 protein.**

A    Representative Western blot of MSTO1 protein expression after both MSTO1-Myc and mtDsRed1-T2A-MSTO1 gene delivery compared to the pcDNA-transfected cell lysate.
B    Subcellular distribution of MSTO1, immunofluorescence visualization was performed with anti-MSTO1 antibody in HeLa cells, which were transfected with mtDsRed1-T2A-MSTO1 vector 24 h before fixation.
C    Localization of MSTO1 protein after 24-h cotransfection with an OMM-targeted GFP (OMP25-GFP) and MSTO-cMyc vectors. The intact (upper part) and permeabilized (lower part) cells were immunostained with anti-MSTO1 or anti-cMyc antibodies.
D    Percentage of MSTO1 colocalization with the mitochondria in intact cells (imaged cells *N* = 30) (mean ± SEM).
E    Percentage of MSTO1 colocalization with the mitochondria in permeabilized cells (imaged cells *N* = 30) (mean ± SEM).
F    Comparative Western blotting of the MSTO1 expression in total cell lysate and isolated mitochondria fraction in HeLa cells.

Source data are available online for this figure.

a role in mitochondrial morphogenesis and quality control by supporting mitochondrial fusion. MSTO1 likely interacts with the mitochondrial fusion machinery as a soluble factor at the cytoplasm–OMM interface.

Sequencing of DNA isolated from the blood (whole exome and Sanger sequencing) assumed that Val8Met mutation in the *MSTO1* gene is responsible for the clinical symptoms in the present family. The cDNA sequencing confirmed the heterozygous V8M, while in the genomic DNA no exon loss or multiplication was found in this gene. Thus, the Hungarian patients seem to have one impaired and one normal MSTO1 allele, which result in a partial decrease in

MSTO1 mRNA and protein levels. Based on combination of several criteria, no other known pathogenic mutation is segregated in this family, but we cannot exclude that partial loss of MSTO1 comes together with some other factors to cause the patients clinical picture. Notably, several mitochondria-associated protein disturbances (e.g., *POLG1, RRM2B, OPA1, MFN2*) can be inherited in both autosomal dominant and recessive manner. Based on previous observation, the autosomal recessive forms usually associated with severe, early-onset phenotypes, while in the autosomal dominant forms linked to milder, adult-onset manifestations (Tyynismaa *et al*, 2009). Loss-of-function mutations are usually associated with a

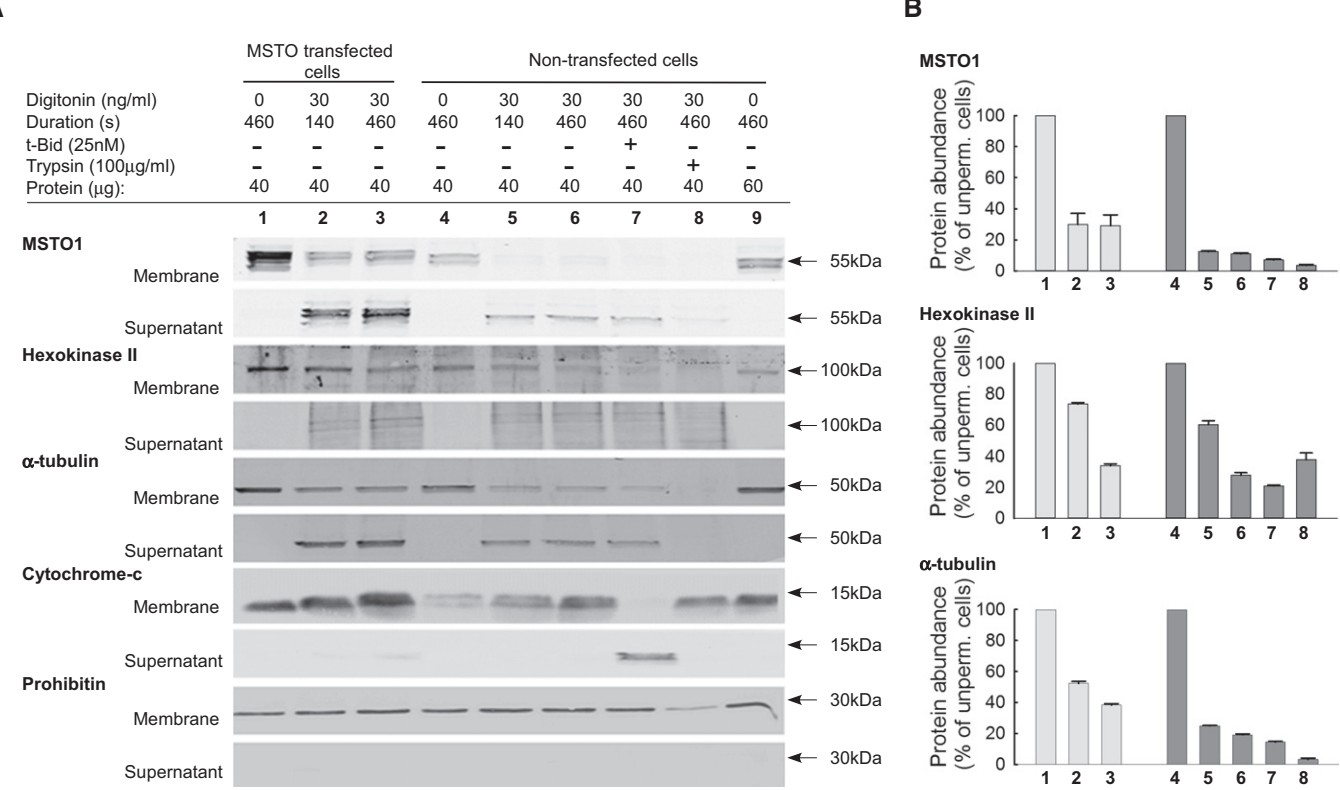

**Figure 8.   MSTO1 protein release after plasma membrane permeabilization.**

A    Representative Western blots of MSTO1, hexokinase II, α-tubulin, and cytochrome c in membrane and supernatant fractions of MSTO1-cMyc-transfected and non-transfected cells.
B    Protein release of MSTO1, hexokinase II, α-tubulin after plasma membrane permeabilization (values are normalized to the intact cells). *N* = 3 (mean ± SEM).

Source data are available online for this figure.

lower amount of the expressed proteins, but depending on its mono- or biallelic features, the remaining protein level could be different. For example in the case of OPA1, the autosomal dominant mutations are associated with reduced OPA1 expression (Bonifert et al, 2014), while for the recessive form almost absent protein expression was found (Spiegel et al, 2016).

Human Misato (MSTO1) was described as an evolutionarily conserved OMM protein, and it has been implicated in the control of mitochondrial morphology (Miklos et al, 1997; Gurvitz et al, 2002; Kimura & Okano, 2007). Our experiments extended this observation by demonstrating that a decrease in MSTO1 protein in patient fibroblasts as well as in HeLa cells upon MSTO1 acute silencing was associated with a decrease in mitochondrial fusion activity specifically (Figs 2–5). The cause–effect relationship between MSTO1 decrease and fusion inhibition was validated by genetic rescue studies in the patient fibroblasts (Fig 4). Since the decrease in mitochondrial fusion was observed upon 72-h silencing, it unlikely results from a complex adaptation process. Furthermore, we have shown that at the same time point, no impairment in $\Delta\Psi_m$ or $Ca^{2+}$ handling or basal respiration was noticeable (Fig 6). Thus, fusion inhibition is not secondary to an impairment of bioenergetics when MSTO1 is decreased.

In both fusion and fission, the key factors are the large GTPase enzymes belonging to the large dynamin superfamily; fusion is mainly regulated by MFN1, MFN2, and OPA1 (Meeusen et al, 2006; Song et al, 2009), while fission requires mainly DRP1 (Otera et al, 2010; Loson et al, 2013). However, the protein expression of these proteins was unaltered in MSTO1 fibroblasts (Fig 3F) as well as in HeLa cells after MSTO1 silencing (Fig 5A).

MSTO1 protein was originally described as an OMM protein; it was co-localized with cytochrome c and MitoTracker Orange in HeLa cells (Kimura & Okano, 2007), and thus, it could be an OMM fusion mediating factor. However, in our experiments, both overexpressed (Fig 7) and endogenous (Fig 8) MSTO1 showed cytoplasmic localization. Since the majority of endogenous MSTO1 was promptly released upon plasma membrane permeabilization with a small fraction remaining associated with membranes (Fig 8), including the mitochondria (Fig 7), MSTO1 likely is a soluble cytoplasmic protein that might interact with OMM components. Thus, MSTO1 might serve as a cytoplasmic regulator of the OMM fusion machinery. Interestingly, partial depletion of MSTO1 is sufficient to impair fusion (Figs 1 and 5), indicating that MSTO1 might be quantitatively required for normal mitochondrial fusion activity. However, because of the lack of direct evidence for an interaction of MSTO1 with a fusion protein, we cannot completely exclude the possibility that the decrease in MSTO1 induces a generic cellular stress response. The main argument against this mechanism is that many aspects of cell and mitochondrial function, including motility and bioenergetics remained unaltered in the MSTO1-deficient cells. In any case, further mechanistic studies will be needed to determine the exact mechanism of the fusion supporting role of MSTO1.

Mitochondrial fusion is central for the maintenance of mtDNA and function (Ono et al, 2001; Chan, 2006; Chen et al, 2010). It is sensible that short-term depletion of MSTO1 did not impair bioenergetics but on the long run in the patients, MSTO1 depletion is expected to cause some impairment in mitochondrial function. Notably, recent results indicate that impaired trafficking of functional mitochondria can also cause neuronal and other

tissue's dysfunction (Nguyen et al, 2014) and mitochondrial movements are affected by the fusion dynamics (Liu et al, 2009). However, MSTO1 also has an involvement in cell division that might be also relevant for some congenital malformations of the proband. DML1 and Misato might be co-opted into a role in mtDNA inheritance in yeast, and into a cell division-related mechanism in flies. The mutations of DML1 or Misato could inhibit kinetochore-driven microtubules growth, associated with a strong mitotic defect; and it has an additional function in the partitioning of mitochondrial organelle itself, or in segregation of chromosomes (Miklos et al, 1997; Gurvitz et al, 2002; Mottier-Pavie et al, 2011).

In conclusion, our findings indicate that MSTO1 has a role in mitochondrial morphogenesis and maintenance by specifically supporting mitochondrial fusion. In contrast to previous findings, MSTO1 likely interacts with the mitochondrial fusion machinery as a soluble factor at the cytoplasm–OMM interface. Since a loss-of-function mutation in MSTO1 is associated with impaired mitochondrial dynamics and several symptoms characteristic to mitochondrial disease patients, MSTO1 might be a factor in diseases that have a mitochondrial component.

# Materials and Methods

### Patient selection and clinical investigations

The patients and controls were collected from the NEPSY Biobank of the Institute of Genomic Medicine and Rare Disorders, Semmelweis University (Molnar & Bencsik, 2006). Written informed consent was obtained from each individual before sample collection and molecular genetic testing. The study was approved by the National Ethical Committee (37/2014 TUKEB). Molecular genetic analysis was performed for diagnostic purposes in all investigated patients.

Detailed neurological, cardiology, psychiatric examinations, and laboratory investigations were performed. Skeletal muscle biopsy was obtained for light and electron microscopic examination using standard routine staining. Electroneurography and electromyography were performed by standard techniques (Dantech Keypoint, Denmark).

### Molecular genetic studies

DNA was extracted from blood and skeletal muscle using QIAamp DNA blood and tissue kits, according to the instructions of manufacturer (QIAGEN, Hilden, Germany).

The total coding region of mtDNA was screened by Sanger sequencing using ABI Prism 3500 DNA Sequencer (Applied Biosystems, Foster City, USA); the obtained sequences were compared with the mitomap databases using NCBI's Blast® application, while mtDNA deletion was investigated with long PCRs using Phusion High-Fidelity DNA Polymerase (Finnzyme, Vanta, Finland).

Exome Capture was performed in two steps according to the manufacturer's protocol. Genomic DNA library preparation was performed by using TruSeq® DNA Sample Prep Kit v2-Set A (Illumina), followed by NimbleGen SeqCap EZ Human Exome Library

v3.0 Kit exome enrichment (Roche). First pre-capture genomic library was prepared using 1 μg of highly purified genomic DNA. Crude pre-captured genomic library was analyzed by Agilent 2100 Bioanalyzer 1000 DNA chip to assess library quality. Next, during exome enrichment step, individual pre-captured libraries were hybridized to biotinylated NimbleGen SeqCap EZ Human Exome Library for 68–70 h at 47°C. Hybridized libraries were captured by magnetic Dynabeads MyOne Streptavidin C1 (Thermo Fisher Scientific). Streptavidin beads were washed, and the captured DNA was eluted. Eluted DNA was PCR amplified by 18 cycles. Crude target-captured libraries were gel purified on 2% E-Gel (Life Technologies) and cleaned up with QIAquick Gel Extraction Kit (QIAgen, Hilden, Germany). Target-captured, gel-purified libraries were analyzed by Agilent 2100 Bioanalyzer 1000 DNA chip to assess libraries quality. The final library concentration was determined by manual integration of molecular profile by Agilent 2100 Bioanalyzer 1000 DNA chip. Illumina adaptor-specific qPCR was also performed for each library to measure the library template concentration containing adaptor sequences on both ends which will subsequently form clusters on the Illumina flow cell. All library pools were sequenced on the HiScanSQ Illumina sequencing platform, using 2 × 95-bp pair-end sequencing protocol, with an extra 9-bp index sequencing run. Final exome-captured sequencing libraries were diluted to 10 nM. Reads were filtered according to the Q30 standard. 95-bp paired-reads were aligned to the human reference genome (hg19). The alignment was performed using Burrows–Wheeler aligner (BWA) software. For variation calling, Samtools software was used (Li *et al*, 2009; Li, 2011). After identification of variants, we focused only on non-synonymous variants, splice acceptor and donor site mutations, and short, frame shift coding insertions or deletions (indel) using Genomes Management Application (GEM.app) software (Department of Human Genetics, University of Miami Health System, and Miami, FL, USA).

The WES filtering procedure was the following: GATK (QUAL) > 50, GQ > 40, RD > 4 quality scores; autosomal dominant inheritance; only the missense, nonsense, indels, and splice-site rare variants with MAF < 0.01 were selected. All SNVs higher than three occurrences in GEMapp were excluded.

For crosschecking, the mitochondrial genetic alterations were investigated by targeted screening for both autosomal dominant and recessive inheritance performed in genes, which is responsible for mitochondrial function. The rare variants were filtered out based on mitochondrial gene function and disease association used a gene list, which was created by UNIPROT, NextProt, MitoMiner, and NCBI databases.

As large-scale genomic data are not available from the Hungarian population, a mutation with low minor allele frequency may be population specific. Finally, we prioritized mutations based on consequence. Exonic frameshift and stop mutations were considered as damaging. Missense mutations were prioritized, which was based on the protein prediction score annotations given by ANNOVAR (Wang *et al*, 2010). Further narrowing was based on protein prediction scores (Polyphen2 score > 0.5, Mutation taster: disease causing, SIFT < 0.1, PHASTCONS > 0.5; GERP > 3). Using ACMG guideline (Richards *et al*, 2015), the pathogenic and likely pathogenic variants were confirmed by Sanger sequencing. After identification of candidate alterations, the segregation analysis was performed in all affected family members.

For the validation of the exome capture result *COL5A1* Exon 3, total coding region of *MSTO1* from genomic and cDNA, *RELN* Exon 62, *RYR2* Exon 16, were analyzed with bidirectional Sanger sequencing (the primers listed in Table EV2). The obtained sequences were compared with the human reference genome (ENST00000371817, NM_000093.4; ENST00000245564, NM_018116.3; ENST00000343529, NM_173054.2; ENST00000366574, NM_001035.2) using NCBI's Blast® application.

The analysis of copy number variation of all exons of *MSTO1* gene was performed by quantitative real-time RT–PCR with SYBG methodology in the real-time PCR StepOnePlus system according the manufacturer's instructions (Life Technology). For quantification, the ddCt method was used.

### RNA isolation and RT–PCR

Total cellular RNA was isolated using by RNeasy Plus Mini Kit, according to the instructions of manufacturer (QIAgen, Hilden, Germany). cDNA was synthesized by high Sensitive cDNA reverse transcriptase kit (Life Technology) according to the manufacturer's instructions. For measuring mRNA level, RT–PCR products were analyzed by quantitative real-time RT–PCR with TaqMan Gene Expression Assays of MSTO1 genes (Hs01094406_g1), and GAPDH (Hs02758991_g1) as endogenous control (Life Technology). All PCRs were performed in the real-time PCR StepOnePlus system. For quantification of gene expression level, the ddCt method was used.

### Creation of a Mito-DsRed1-T2A-MSTO1 construct

Human MSTO1-Myc was purchased from OriGene (clone RC200136). To create the mtDsRed1-T2A-MSTO1 construct, first the sequences of the mtDsRed1 (from the pDsRed1-Mito plasmid, Clontech) and the human MSTO1 (from the cDNA clone RC200136, OriGene) were cloned into the pEGFP-N1 plasmid (Clontech) using the NheI and NotI restriction sites. Next, the sequence of the viral T2A peptide was inserted between Mito-DsRed1 and MSTO1 resulting the cleavage of the single mRNA-coded polypeptide chain into two pieces during translation (PMID: 22357943). Transfection was performed using Lipofectamine 2000 (Life Technologies) with 0.7–4 μg of plasmid DNA in serum-free culture medium for 4 h.

### Cell lines, cultures, and transfections with siRNA and plasmid DNA

Fibroblasts were generated from a 6-mm skin punch biopsy taken under local anesthetic of healthy control individuals (Ctrl1: 35-year-old male, Ctrl2: 25-year-old female, Ctrl3: 57-year-old male) and patients carrying a c.22 G>A, p.Val8Met mutation in *MSTO1* gene (MSTO P1, MSTO P2). Cells were grown in Dulbecco's modified Eagle's medium (DMEM), with 4.5 g/l glucose supplemented with 10% fetal bovine serum, 2 mM glutamate, 100 units/ml penicillin/streptomycin (all from GIBCO, Life Technology), and 110 mg/l pyruvate (Sigma). All cells were grown at 37°C in a humidified 5% $CO_2$ incubator. HeLa cells (ATCC) were grown in the above-described medium without pyruvate.

Before silencing, the cells were plated in antibiotic-free medium, and the samples were silenced in serum-free culture medium with MSTO1-specific (s30301, Life Technologies) or scrambled siRNAs

(Silencer® Negative Control #1 siRNA, Life Technologies) (100 nM), using Oligofectamine (Life Technologies) for 72 h. The efficacy of silencing was verified with Western blotting. For evaluation of mitochondrial morphology and fusion dynamics, the cells were transfected with mtDsRed1 or mtDsRed1-T2A-MSTO1 and mtPA-GFP plasmid DNAs 24 h before the experiment.

## Western blotting

For Western blotting, 40 μg of proteins was loaded into each lane of a 12% SDS–PAGE and electrophoretically transferred to nitrocellulose filters. Membranes were blocked with blocking buffer (LI-COR Biosciences), followed by overnight incubation with primary antibodies (Table EV3). Bound antibodies were visualized with IRDye 800 secondary antibodies (LI-COR Biosciences). Quantification of protein abundance was done by ImageJ software and abundance, and the relevant bands were normalized to loading controls (mtHSP70 or prohibitin).

## Mitochondria preparation

The cells were trypsinized, then trypsin was neutralized with 2% BSA, and the cells were washed with Na-HEPES–EGTA. The pellet was resuspended in a hyposmotic intracellular medium (24 mM KCl, 0.2 mM $KH_2PO_4$, 2 mM NaCl, 4 mM HEPES–Tris, 200 μM EGTA, 1 μg/ml of antipain, leupeptin, and pepstatin, 5 mM $MgCl_2$, 100 μM PMSF) and centrifuged at $1,000 \times g$. The supernatant was homogenized with cell cracker (18 nm clearance, 15×), and the isoosmolarity was restored by addition 125 mM sucrose, 120 mM KCl, 1 mM $KH_2PO_4$, 10 mM NaCl, 20 mM HEPES–Tris, 200 μM EGTA, 100 μM PMSF before centrifugation at $1,000 \times g$. The supernatant was centrifuged at $10,000 \times g$, and the final pellet (mitochondrial fraction) was resuspended in protein lysis buffer (RIPA, 1 μg/ml antipain, leupeptin, and pepstatin, and 100 μM PMSF). All steps were performed at 4°C.

## Immunocytochemistry

The adherent cells were fixed with 3% paraformaldehyde in fixation buffer at 37°C and were immunostained with the anti-MSTO1 (OriGene) or anti-cMyc (Santa Cruz Biotechnology) primary antibodies. The secondary antibodies were labeled with Alexa-488 and Alexa-546 (Thermo Fisher Scientific). After staining, the coverslips were post-fixed with 3% paraformaldehyde. In some cases, the cells were permeabilized before the immunostaining. The cells were washed with prewarmed (37°C) Na-HEPES–EGTA and permeabilized with 30 μg/ml digitonin in ICM (120 mM KCl, 10 mM NaCl, 1 mM $KH_2PO_4$, 20 mM Tris-HEPES, pH 7.2, 1 μg/ml antipain, leupeptin, and pepstatin, 0.4 mM KOH, and 2 μM thapsigargin). Image acquisition (512 × 512 pixels) was performed using 488 nm and 561 nm laser lines of an LSM780 confocal laser-scanning microscope [63×/1.4 NA or 40×/1.4 NA, PlanApo (Carl Zeiss)]. Image analysis was done using either Spectralyzer (custom designed) or ImageJ software. The percentage of colocalization was calculated with an ImageJ macro application.

## Live cell imaging

Imaging was performed as previously described (Eisner *et al*, 2014; Weaver *et al*, 2014). Briefly, cells were incubated in a 0.25% BSA

containing extracellular medium (ECM) consisting of 121 mM NaCl, 5 mM $NaHCO_3$, 4.7 mM KCl, 1.2 mM $KH_2PO_4$, 1.2 mM $MgSO_4$, 2 mM $CaCl_2$, 10 mM glucose, and 10 mM Na-HEPES, pH 7.4, at 37°C. Recordings of mtPA-GFP and mtDsRed (512 × 512 pixels) were performed using 488 and 561 nm laser lines at 0.25 /s data acquisition frequency using the LSM780 microscope. To photoactivate PA-GFP in 2P mode, a pulsed laser system (760 nm, Chameleon; Coherent, Inc.) was applied. Image analysis was done using either Spectralyzer or Zen2010.

## Analysis of mitochondrial matrix continuity

Mitochondrial matrix continuity was evaluated by two different approaches. Spreading of PA-GFP from the area of photoactivation was evaluated by masking the 5 × 5 μm areas and quantifying the time-dependent decay in the fluorescence intensity. The images were thresholded to reach a 0.1% noise-to-signal ratio. Then, the pixels of the photoactivated areas were subtracted and the PA-GFP-positive pixels were counted. Image analysis was performed in Spectralyzer imaging software or in Zen2010. Only cells with the same number of photoactivated areas and no focus loss were compared.

## Analysis of GFP-only pixel loss

Automated fusion analysis was accomplished by tracking the area of GFP-only fluorescence after the simultaneous photoactivation/photobleaching of PA-GFP and DsRed/RFP. Briefly, in ImageJ, a rough region of interest was drawn around the entire cell and the period after photoactivation selected; after "despeckle" filtering, a single threshold was set for each image series (PA-GFP and RFP) using the Otsu method. The thresholded RFP series was subtracted from the GFP to yield the GFP-only areas. The number of green-only pixels over time was fit by a single exponential (y = ae-bx), and the time constant (b) (or half-time ln(2)/b) of the exponential decay was used as the measure of fusion. This measure was verified to correlate with the number of fusions by manual counting in recordings of H9c2 cells ($R^2 = 0.698$, $P < 0.0001$, $n = 31$).

## Fluorometric measurements of mitochondrial membrane potential and Ca²⁺ uptake

The cells (1.6 mg/ml) were suspended in an ICM supplemented with 2 mM MgATP, 1 mM malate, 1 mM pyruvate, 2 μM thapsigargin, 1 μM FuraFF/FA (TEFLabs), and 2 μM TMRE (Invitrogen). The extramitochondrial $Ca^{2+}$ concentration was measured using FuraFF, whereas the deltapsi was assessed with TMRE. Fluorescence was monitored in a multiwavelength excitation/dual wavelength emission fluorimeter (DeltaRAM; Photon Technology International). FuraFF and TMRE fluorescence were recorded simultaneously using 340–380 nm excitation and 500 nm emission, and 545 nm excitation and 580 nm emission, respectively. Calibration of the FuraFF signal in terms of μM $[Ca^{2+}]$ was carried out at the end of each measurement by adding 2 mM $CaCl_2$, followed by 10 mM EGTA/Tris, pH 8.5. TMRE time courses were normalized to the fluorescence obtained, in the fully depolarized condition upon addition of an uncoupler, 5 μM FCCP.

To monitor dynamically the MSTO1 protein distribution, the plasma membrane was permeabilized in suspensions of HeLa cells

and human fibroblasts. The HeLa cells were effectively permeabilized with digitonin (30 μg/ml), while the primary fibroblast resisted to digitonin up to 60 μg/ml, but undergone plasma membrane permeabilization when exposed to saponin (40 μg/ml). The amount of digitonin and saponin required was determined by trypan blue uptake (> 95% of the cells). The samples were continually stirred in a fluorometer cuvette. At specific time points, the samples were centrifuged at $10,000 \times g$ to separate the membrane proteins (pellet) from the cytoplasmic fraction (supernatant). After immunoblotting of the membrane and cytosolic fractions, the protein abundances were normalized to total cellular protein.

### Measurement of respiratory function

HeLa cells were treated with siRNA against MSTO, or scrambled control siRNA, for 72 h, as already described. The siRNA against MSTO resulted in a ~50% (48% ± 9%, mean ± SEM) knockdown of MSTO protein. To perform mitochondrial $O_2$ consumption rate ($JO_2$) experiments using the Seahorse XF24 analyzer, cells were seeded into custom plates at 28,000/well the day before the experiment and cultured overnight. To measure $JO_2$, culture media was replaced by assay media (DMEM plus 25 mM glucose and 4 mM glutamine as substrates). Cells were then incubated at 37°C, zero $CO_2$ for 45 min; then, the experiment was started. $JO_2$ measurements comprised repeated cycles of a 2-min mix, a 2-min wait, and a 2-min measurement. To measure maximal leak-dependent $JO_2$, oligomycin (2.5 μg/ml; ATP synthase inhibitor) was injected. To measure maximal electron transport chain activity under the prevailing substrate conditions, the chemical uncoupler FCCP (250 nM) was injected. This [FCCP] represents the optimal [FCCP] based on studies in which these cells were exposed to different [FCCP]. To measure non-mitochondrial $O_2$ consumption rate, antimycin A (1 μM; complex III inhibitor) was injected. Mitochondrial $O_2$ consumption (i.e., $JO_2$) was calculated as the rate prior to antimycin minus the rate post-antimycin injection. At the conclusion of each experiment, protein was lysed *in situ* and total protein was measured by the BCA method. $JO_2$ data were normalized by total protein/well.

### Statistics

The data are shown as the mean ± SEM of cells from at least three independent cultures unless it is specified differently. Significance of differences was calculated by one-way ANOVA analysis or Student's *t*-test.

**Expanded View** for this article is available online.

### Acknowledgements

We are grateful to the patients and their clinicians and healthy volunteers for providing samples. Thanks to Stephan Züchner for access to GEM.app software; to Heidi McBride and Richard J. Youle for reagents, Gyorgy Csordas, David M. Booth, Melanie Paillard, Valentina Debattisti, Tunde Golenar, and Gyorgy Bathori for helpful discussions; and to Gyorgyi Bathori, Marianna Marko, and Monika Sary for their technical help. This study was supported by the Hungarian Brain Research Program (KTIA_13_NAP-A-III/6 to V.A-V.) and a Schizo2010 Consortium grant to M.M.J., an HAESF Senior Leadership Program Fellowship to A.G., and an NIH grant (DK51525) to G.H.

### The paper explained

#### Problem
Genetic impairments of mitochondria-associated proteins are potential contributors in a range of human disorders, but validation of their variations and determination of their specific functions is difficult.

#### Results
This work reveals the first dominant mutation in MSTO1 associated with multisystemic disease in a Hungarian family. Furthermore, the role of MSTO1 in shaping of mitochondria via regulation of mitochondrial fusion is directly demonstrated. At variance with previous work, MSTO1 is shown to be a soluble cytoplasmic protein that can interact with the fusion proteins of the outer mitochondrial membrane.

#### Impact
Thanks to a multidisciplinary collaboration among clinicians, geneticists, and cell biologists, MSTO1 is found to be a new disease-associated gene responsible for a rare mitochondrial multisystemic disorder.

## Author contributions

AG, PB, TG, AH, and LN performed the genetic analysis. MJM supervised all the genetic work, diagnosed the family, and performed the clinical examinations. AG performed all the cell biology studies with the guidance of SN and SKJ. ELS designed and carried out the respiratory function measurements. DW created software for image analysis and PV generated new reagents. GH designed and supervised the cell biology studies. AG, MJM, and GH wrote the manuscript. All the authors critically revised the manuscript for intellectual content.

## Conflict of interest

The authors declare that they have no conflict of interest.

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
