## [Review Process File · EMBO Molecular Medicine]

MSTO1 is a cytoplasmic pro-mitochondrial fusion protein, whose mutation induces myopathy and ataxia in humans

Aniko Gal, Peter Balicza, David Weaver, Shamim Naghdi, Suresh K. Joseph, Péter Várnai, Tibor Gyuris, Attila Horváth, Laszlo Nagy, Erin L. Seifert, Maria Judit Molnar, György Hajnóczky

Corresponding author: Gyorgy Hajnoczky, Thomas Jefferson University

Review timeline:

Submission date:	13 September 2016
Editorial Decision:	07 November 2016
Revision received:	08 February 2017
Editorial Decision:	20 March 2017
Revision received:	13 April 2017
Accepted:	21 April 2017

Transaction Report:

Editor: Roberto Buccione

1st Editorial Decision

07 November 2016

Thank you for the submission of your manuscript to EMBO Molecular Medicine. We are very sorry that it has taken so long to get back to you on your manuscript. In this case we experienced unusual difficulties in securing three willing and appropriate reviewers, possibly also due to the challenge of dealing with two back-to-back submissions. As a further delay cannot be justified I have decided to proceed based on the two available consistent evaluations.

In addition to the above, further extensive internal discussion was required to reach a final decision.

As you will see, while Reviewer 2 is largely positive, Reviewer 1 is more reserved. They both raise a number of important and partially overlapping issues that require your action. I will not go into much detail, as their comments are quite clear.

After reviewer cross-commenting and the internal discussion, it was agreed that while I will not be asking you to provide further experimentation concerning reviewer 1's point 5 (although any further data along those lines would be welcome), I must ask you to take action on the remaining concerns by both reviewers. We would consider changing the manuscript format from a "Report" to a "Research Article".

In conclusion, while publication of the paper cannot be considered at this stage, we would be pleased to consider a revised submission, with the understanding that the Reviewers' concerns must be fully addressed as detailed above including with additional experimental data where appropriate and that acceptance of the manuscript will entail a second round of review. Please note that your manuscript will be from now on handled independently from the accompanying submission.

Please note that it is EMBO Molecular Medicine policy to allow a single round of revision only and that, therefore, acceptance or rejection of the manuscript will depend on the completeness of your responses included in the next, final version of the manuscript.

As you know, EMBO Molecular Medicine has a "scooping protection" policy, whereby similar findings that are published by others during review or revision are not a criterion for rejection. However, I do ask you to get in touch with us after three months if you have not completed your revision, to update us on the status. Please also contact us as soon as possible if similar work is published elsewhere.

Please note that EMBO Molecular Medicine now requires a complete author checklist (<http://embomolmed.embopress.org/authorguide#editorial3>) to be submitted with all revised manuscripts. Provision of the author checklist is mandatory at revision stage; The checklist is designed to enhance and standardize reporting of key information in research papers and to support reanalysis and repetition of experiments by the community. The list covers key information for figure panels and captions and focuses on statistics, the reporting of reagents, animal models and human subject-derived data, as well as guidance to optimise data accessibility. In this case, the author checklist is especially relevant as, in addition to the concerns on the clinical features of the TMA, I note that both reviewers have reservations on your presentation of statistics information. The Author checklist will be published alongside the paper, in case of acceptance, within the transparent review process file

Please note that we now mandate that all corresponding authors list an ORCID digital identifier. You may do so through our web platform upon submission and the procedure takes <90 seconds to complete. We also encourage co-authors to supply an ORCID identifier, which will be linked to their name for unambiguous name identification.

I look forward to seeing a revised form of your manuscript as soon as possible.

***** Reviewer's comments *****

Referee #1 (Remarks):

The authors report a heterozygous missense mutation in MSTO1 in four members in a single pedigree with complex neurological phenotypes. The MSTO1 mRNA and protein was reduced in patient fibroblasts. However, the activities of OXPHOS enzymes and respiration were normal. The distribution of mitochondrial morphologies differed between patient and control fibroblasts. A number of experiments with labeled mitochondria and a photoactivatable GFP showed that the number of fusion events was reduced in patient cells as was mitochondrial connectivity. This could be rescued by expression of the wild-type cDNA. Suppression of MSTO1 phenocopied the patient cells. Although MSTO1 was previously reported to be mitochondrial, immunofluorescence studies showed it to be primarily cytoplasmic. The bioenergetics status of patient mitochondria was unaltered based on measurements of mitochondrial membrane potential and calcium uptake. The authors conclude that MSTO1 is a cytosolic protein that likely interacts with the fusion machinery to promote mitochondrial fusion and that this is necessary for mitochondrial quality control.

I have the following comments/questions:

(1) It is not at all clear to me how they filtered the WES results to get down to the rather small number of variants they considered. Not clear whether they only considered a dominant model, or whether they also looked for homozygous or compound heterozygous mutations that might produce loss of function. This needs to be explained in much more detail.

(2) Are the very small regions of homology of MSTO1 with MFN1,2 statistically significant?

(3) The pattern of mitochondrial morphology in the patient cells shown here is quite different than that described in the companion paper, in which all mitochondria were shown to be fragmented, and I guess this represents loss of function vs. haplo-insufficiency. The largest difference between controls and patients in this study lies between the relative proportions of normal vs. partially

fragmented mitochondria, and I must say that seems to me a rather tough call to me based on their representative images. (Admittedly, the patient cells do have a small percentage of fragmented mitochondria that do not appear in control cells). In any case the morphological differences seem to me to be rather subtle.

(4) While I find the results of the fusion experiments clear, the effects do not seem all that large to me (but correct me if I am wrong). It would have been very helpful to have a positive control ie a bone fide defective fusion protein, so one could compare altered fusion parameters with the differences reported here

(5) They conclude that MSTO1 is somehow involved with the mitochondrial fusion machinery but in fact provide no direct evidence of this, like recruitment to the mitochondrial surface under conditions promoting fusion, chemical crosslinking to outer mitochondrial membrane proteins etc. A variety of different cellular stresses that have nothing to do with the fission/fusion machinery per se can result in alterations in mitochondrial morphology, and it is entirely possible that haplo-insufficiency of MSTO1, is one such stress.

To summarize, I need to be convinced that they have ruled out all other potentially damaging mutations in other genes from the WES data set. I think it premature to conclude that MSTO1 plays a direct role in fusion without some evidence that it influences or interacts in some way with the known components of the fusion machinery. Finally, I must say that I am not really aware of any direct evidence that mitochondrial fusion is necessary for quality control.

Referee #2 (Comments on Novelty/Model System):

This is an outstanding manuscript

Referee #2 (Remarks):

This is an outstanding report on the discovery of a novel pro-fusion molecule. The study is robust and include the state of the art fusion and dynamics assays. There is proper analysis of over expression which supports the study hypothesis. Moreover the authors provide evidence that transient silencing also results in reduced fusion activity confirming that this phenotype is unlikely to be a long term compensation. Major points:

1. Authors should address the possibility that MSTO is affecting mitochondrial motility and then indirectly affecting mitochondrial fusion and fragmentation. This group has previous published that mitochondrial motility is a key dominant of mitochondrial fusion. As such, it is essential to address the possibility of motility as an upstream mechanism.
2. There is no address of the effect of MSTO on mitochondrial bioenergetics function (respiratory function). Changes in membrane potential are observed in uncompensated conditions and are thus the most insensitive parameter for bioenergetics defect. At least inclusion of minimal analysis of maximal respiratory capacity and proton leak of the knockdown model is essential.
3. Section on MSTO solubility is unclear and requires explanation of the reasoning behind the experiments

1st Revision - authors' response

08 February 2017

Answer to the Reviewers' questions:

We appreciate the positive comments and constructive criticism of both Reviewers, which helped us to improve our study.

Referee #1

(1) It is not at all clear to me how they filtered the WES results to get down to the rather small number of variants they considered. Not clear whether they only considered a dominant model, or

whether they also looked for homozygous or compound heterozygous mutations that might produce loss of function. This needs to be explained in much more detail.

In the WES analysis we followed the below listed main steps.

1. First, we set the following quality scores: GATK (QUAL) > 50, GQ>40, RD>4.
2. We used the GEMapp software to filter the autosomal dominant inheritance, the missense, nonsense, indels and splice site variants were selected with a MAF of <0.01.
3. All SNVs with higher score than three of GEMapp were excluded.
4. Next filtering was performed with in silico prediction software (Polyphen2 score>0.5, Mutation taster: disease causing, SIFT <0.1, PASTCONS>0.5; GERP>3)
5. We also excluded the genes with unknown function or without any mitochondrial or central nervous system involvement.
6. Although the family has autosomal dominant inherited symptoms the homozygous and compound heterozygous rare variants were also filtered to all mitochondrial associated genes (1050 genes, based on Uniprot, Nextprot, Mitominer and NCBI databases). We did not find any of them with the above mentioned filtering criteria.
7. Finally, the confirmation and segregation analyses of four remaining pathogenic or likely pathogenic alterations were performed by Sanger sequencing. Among them only the MSTO1 V8M mutation was present in all affected family members. For the stratification of variants, the ACMG guideline was used.

These steps were added to the text in the results and methods sections. See in page 6 para 1 and page 17.

(2) Are the very small regions of homology of MSTO1 with MFN1,2 statistically significant?

To determine the significance of the MSTO1-Mfn1 alignment, BLAST alignments using randomly scrambled versions of each protein were performed with the default parameters for multiple sequence alignment (word size 3, BLOSUM62 matrix). Out of 100 alignments Mfn1 and scrambled MSTO1s and 100 alignments of MSTO1 and scrambled Mfn1s, 17 (8.5%) aligned segments with an equal or higher bit-score were found. Thus, the homology is not statistically significant in and of itself, though we would note the strong conservation across species of the acidic amino acids in MSTO1 and mitofusins.. This has been included in the revised text (pg7 para3).

(3) The pattern of mitochondrial morphology in the patient cells shown here is quite different than that described in the companion paper, in which all mitochondria were shown to be fragmented, and I guess this represents loss of function vs. haplo-insufficiency. The largest difference between controls and patients in this study lies between the relative proportions of normal vs. partially fragmented mitochondria, and I must say that seems to me a rather tough call to me based on their representative images. (Admittedly, the patient cells do have a small percentage of fragmented mitochondria that do not appear in control cells). In any case the morphological differences seem to me to be rather subtle.

The MSTO1-deficient cells (patient cells and siRNA-treated cells) showed a consistent increase in mitochondrial fragmentation and condensation, which was reversed by MSTO1 rescue (in the patient cells). However, such mitochondrial morphology can result from changes in several different processes. Therefore, the specific, fusion activity phenotype we documented is of greater relevance. We would like to point out that the vast majority of the literature uses only representative images of a mitochondria-targeted photo-modifiable fluorescent protein and fluorescence loss kinetics for the region of photo-modification to assess fusion, whereas here, in addition to these, we also performed fusion counting and described for the first time, an un-biased pixel-based fusion quantification algorithm. All the results support a mitochondrial fusion impairment in the MSTO1-deficient cells. Furthermore, in the revised study, we also show that mitochondrial motility was not decreased in the MSTO1 cells, arguing against the possibility that decreased mitochondrial movements would induce the mitochondrial morphology or fusion phenotype. See pg11 para1.

(4) While I find the results of the fusion experiments clear, the effects do not seem all that large to me (but correct me if I am wrong). It would have been very helpful to have a positive control ie a bone fide defective fusion protein, so one could compare altered fusion parameters with the differences reported here

In a previous study, we quantified the effect of 2 different ADOA-associated OPA1 mutations on the mitochondrial fusion activity using the same fusion counting method used in the present study (Eisner et al 2014 J Cell Biol). G300E point mutation in the GTPase domain and D58 deletion of the GTPase domain caused 60-70 % decrease in the fusion activity, whereas the present MSTO1

mutations induced an approx 40% decrease in fusion activity. However, we have also studied an ADOA-associated OPA1 mutation (c.984) that caused a lesser decrease in the fusion activity (30%) than the present MSTO1 mutations. Thus, the MSTO1-associated fusion decrease is in the range of the OPA1-associated ones.

As a positive control, we have cited in the revised ms the results with the OPA1 mutations (see pg8 last para).

(5) They conclude that MSTO1 is somehow involved with the mitochondrial fusion machinery but in fact provide no direct evidence of this, like recruitment to the mitochondrial surface under conditions promoting fusion, chemical crosslinking to outer mitochondrial membrane proteins etc. A variety of different cellular stresses that have nothing to do with the fission/fusion machinery per se can result in alterations in mitochondrial morphology, and it is entirely possible that haplo-insufficiency of MSTO1, is one such stress.

A decrease in the fusion activity was observed when the cells were maintained under optimal conditions and no apparent signs of cellular discomfort were noticed in the MSTO1 cells. In the revised ms we also show that the cellular respiration was unaltered. These observations are against the idea that a decrease in the MSTO1 protein induces a cellular stress to suppress fusion activity. We appreciate the Reviewer's suggestions for additional mechanistic studies but with the Editor's approval we would like to pursue these complex experiments as part of a follow up study. Discussion on this point has been added (pg15 para1). "because of the lack of direct evidence for an interaction of MSTO1 with a fusion protein, we can't completely exclude the possibility that the decrease in MSTO1 induces a generic cellular stress response. The main argument against this mechanism is that many aspects of cell and mitochondrial function, including motility and bioenergetics remained unaltered in the MSTO1 deficient cells. In any case, further mechanistic studies will be needed to determine the exact mechanism of the fusion supporting role of MSTO1."

To summarize, I need to be convinced that they have ruled out all other potentially damaging mutations in other genes from the WES data set. I think it premature to conclude that MSTO1 plays a direct role in fusion without some evidence that it influences or interacts in some way with the known components of the fusion machinery. Finally, I must say that I am not really aware of any direct evidence that mitochondrial fusion is necessary for quality control.

Based on the genetic analysis described above in response to (1) we propose that only MSTO1 can explain the mitochondrial impairment and multisystemic symptoms of the investigated family members. Among the other rare variants, no further mitochondrial genes likely influence the mRNA or protein expression levels of MSTO1. We present in this study multiple lines of evidence for the loss of MSTO1 specifically depressing mitochondrial fusion activity, whereas mitochondrial motility, energy metabolism and Ca²⁺ transport remain unaltered. MSTO1 likely targets the outer mitochondrial membrane fusion proteins but the exact mechanism remains to be addressed. The phrase, 'mitochondrial quality control', has been used in a range of contexts. We refer to its broader meaning involving both repair and replenishing. Although, the role of fusion mediated complementation in quality control remains to be further investigated, the study of David Chan's group describing that mitochondrial fusion is required in skeletal muscle for mtDNA stability (Chen et al 2010 Cell) and studies describing reciprocal relationship between fusion and mitophagy/autophagy seem to provide some evidence for the need for fusion in mitochondrial quality control. In any case we have changed the wording of the statement on this (pg15 para2).

Referee #2 (Comments on Novelty/Model System):

This is an outstanding manuscript

Referee #2 (Remarks):

Major points:

1. Authors should address the possibility that MSTO is affecting mitochondrial motility and then indirectly affecting mitochondrial fusion and fragmentation. This group has previously published that mitochondrial motility is a key dominant of mitochondrial fusion. As such, it is essential to address the possibility of motility as an upstream mechanism.

Mitochondrial motility was quantified in the MSTO1 patient fibroblasts (P1 and P2) as well in the Controls (Ctrl1, Ctrl2 and Ctrl3) in side-by-side comparisons (see Fig below and Fig EV3). The results show that P2 had similar mitochondrial movement activity to Ctrl1 and more than Ctrl2, and

P1 was similar to Ctrl3 in terms of mitochondrial motility. Thus, the decrease in mitochondrial fusion cannot be secondary to a decrease in mitochondrial motility in the MSTO1 deficient cells. This information has been included in the ms (pg11 para1).

2. There is no address of the effect of MSTO on mitochondrial bioenergetics function (respiratory function). Changes in membrane potential are observed in uncompensated conditions and are thus the most insensitive parameter for bioenergetics defect. At least inclusion of minimal analysis of maximal respiratory capacity and proton leak of the knockdown model is essential.

For the JO_2 studies in HeLa cells, siRNA against MSTO resulted in $48\% \pm 9\%$ ($n=5$) knockdown of MSTO protein. Partial depletion of MSTO protein was without major effect on basal (Fig below and Fig6C) or leak-dependent JO_2 (not shown), but led to a slight but significant lowering of the maximal electron transport chain activity (Fig below and Fig6C) as determined by injecting the chemical uncoupler FCCP. This is described on pg 11 para 3-4 in the revised ms.

3. Section on MSTO solubility is unclear and requires explanation of the reasoning behind the experiments

The initial goal of the experiments was to clarify the localization of MSTO1, since a previous study localized MSTO1 to the mitochondria whereas our experiments showed a broad cytoplasmic distribution by immunofluorescence. Permeabilization of the plasma membrane resulted in almost complete loss of the immunofluorescence, whereas the mitochondrial markers stayed, indicating that MSTO1 can't be an integral or firmly associated mitochondrial protein. Then it became important to determine how this protein could interact with the mitochondrial fusion proteins which are transmembrane components of the outer or inner mitochondrial membrane. Therefore, we compared the release upon plasma membrane permeabilization of MSTO1 to other cellular proteins, including hexokinase 2 that is a soluble cytoplasmic protein with temporary but functionally relevant association with the outer mitochondrial membrane. Our study confirmed that the vast majority of MSTO1 is released as readily as soluble cytoplasmic proteins, supporting that MSTO1 is a soluble cytoplasmic protein that can directly interact only with the outer membrane fusion machinery from the cytoplasmic side.

We have changed the results to better explain the rationale for these studies (pg12 last para).

2nd Editorial Decision

20 March 2017

Thank you for the submission of your revised manuscript to EMBO Molecular Medicine. I apologise again for the time it has taken to reach a decision on your manuscript due to one reviewer delivering his/her report with considerable delay. I hope that the consequent inevitable frustration is somewhat tempered by the positive outcome.

In fact, as you will see the reviewers are now globally supportive and I am pleased to inform you that we will be able to accept your manuscript pending the following final editorial amendments:

1) As per our Author Guidelines, the description of all reported data that includes statistical testing must state the name of the statistical test used to generate error bars and P values, the number (n) of independent experiments underlying each data point (not replicate measures of one sample), and the actual P value for each test (not merely 'significant' or ' $P < 0.05$ ').

2) We encourage the publication of source data, with the aim of making primary data more accessible and transparent to the reader. Would you be willing to provide a PDF file per figure that contains the original, uncropped and unprocessed scans of all or at least the key gels used in the manuscript and/or source data sets for relevant graphs? The files should be labeled with the appropriate figure/panel number, and in the case of gels, should have molecular weight markers; further annotation may be useful but is not essential. The files will be published online with the article as supplementary "Source Data" files. If you have any questions regarding this just contact me.

3) Please update nomenclature of expanded view material from expanded view table/figure" to "Table/Figure EV1, etc

4) Would you be willing to provide a striking image or visual abstract to illustrate your article. If you decided to do so, please provide a jpeg file 550 px-wide x 400-px high.

Please submit your revised manuscript within two weeks. I look forward to seeing a revised form of your manuscript as soon as possible. The earlier you do so, the sooner I will be able to formally accept for publication.

***** Reviewer's comments *****

Referee #1 (Remarks):

no further comments

Referee #2 (Remarks):

The authors have very properly addressed my comments and suggestions. This is a wonderful study. No further changes are required.

2nd Revision - authors' response

13 April 2017

1) As per our Author Guidelines, the description of all reported data that includes statistical testing must state the name of the statistical test used to generate error bars and P values, the number (n) of independent experiments underlying each data point (not replicate measures of one sample), and the actual P value for each test (not merely 'significant' or 'P < 0.05').

The p values have been checked and were changed to the exact values.

2) We encourage the publication of source data, with the aim of making primary data more accessible and transparent to the reader. Would you be willing to provide a PDF file per figure that contains the original, uncropped and unprocessed scans of all or at least the key gels used in the manuscript and/or source data sets for relevant graphs? The files should be labeled with the appropriate figure/panel number, and in the case of gels, should have molecular weight markers; further annotation may be useful but is not essential. The files will be published online with the article as supplementary "Source Data" files. If you have any questions regarding this just contact me.

We have created a pdf file that contains the source files for each immunoblot and for the Seahorse measurements. *3) Please update nomenclature of expanded view material from expanded view table/figure" to "Table/Figure EV1, etc*

It has been done as requested.

4) Would you be willing to provide a striking image or visual abstract to illustrate your article. If you decided to do so, please provide a jpeg file 550 px-wide x 400-px high.

A visual abstract has been created.

Corresponding Author Name: György Hajnóczky

Journal Submitted to: EMBO Mol Med

Manuscript Number: EMM-2016-07058